# ParaS2S: Benchmarking and Aligning Spoken Language Models for Paralinguistic-aware Speech-to-Speech Interaction

**Shu-wen Yang**[12][†]**, Ming Tu**[2][†]**, Andy T. Liu**[2]**, Xinghua Qu**[2]**, Hung-yi Lee**[1]**, Lu Lu**[2][†]**,
Yuxuan Wang**[2]**, Yonghui Wu**[2]

[1]Graduate Institute of Communication Engineering, National Taiwan University
[2]ByteDance Seed
leo19941227@gmail.com,
{mingtu,lulu.0314}@bytedance.com

## Abstract

Speech-to-Speech (S2S) models have shown promising dialogue capabilities, but their ability to handle paralinguistic cues—such as emotion, tone, and speaker attributes—and to respond appropriately in both content and style remains under-explored. Progress is further hindered by the scarcity of high-quality and expressive demonstrations. To address this, we introduce a novel reinforcement learning (RL) framework for paralinguistic-aware S2S, **ParaS2S**, which evaluates and optimizes both response content and speaking style directly at the waveform level. We first construct **ParaS2SBench**, a benchmark that evaluates the naturalness of input–output pairs in terms of content and speaking style using expressive and challenging queries. For the automatic judge, we propose a PolyTone training strategy and a multi-stage framework, preventing the style hallucination of end-to-end audio LLM judging. Our judge correlates well with human preferences and is scalable, enabling the model to interact and learn from unlabeled speech via RL. Experiments show that existing S2S models fail to respond appropriately to paralinguistic attributes, performing no better than pipeline-based baselines. Our RL approach (**ParaS2SAlign**) achieves an 10% relative improvement in the appropriateness of response content and speaking style on ParaS2SBench over supervised fine-tuning (SFT), surpassing all prior models while requiring substantially fewer paired demonstrations than pure SFT. Our findings highlight the need for a scalable and accurate automatic evaluator for speech-to-speech interaction.[1]

## 1 Introduction

Speech is the most natural medium of communication, conveying not only words but also paralinguistic cues—emotion, tone, and speaker attributes—that jointly shape true intent and guide appropriate responses (Schuller & Batliner, 2013). This interplay of linguistic and paralinguistic signals motivates speech-to-speech (S2S) models (Xu et al., 2025; Huang et al., 2025b; Zeng et al., 2024) for human-like, empathetic interaction beyond text-based dialogue systems (Achiam et al., 2023; Grattafiori et al., 2024).

S2S models show strong dialogue abilities (Fang et al., 2025a;b; Zeng et al., 2024), as seen in Qwen2.5-Omni (Xu et al., 2025) and ChatGPT Voice Mode.[2] Built on LLMs, they preserve reasoning and conversational abilities while adding speech as a new I/O modality, achieving high scores on benchmarks like VoiceBench (Chen et al., 2024) and Llama Questions (Nachmani et al., 2024). Yet most benchmarks focus on question answering (Nachmani et al., 2024), instruction following (Lu et al., 2025), or speech-to-text understanding tasks (Yang et al., 2024; Sakshi et al., 2025b), over-looking paralinguistic-aware dialogue. StyleTalk (Lin et al., 2024a) and VoxDialogue (Cheng et al.,

---

[†]Corresponding authors.
[1] Project page and demo: https://paras2sbench.github.io/
[2]https://openai.com/index/chatgpt-can-now-see-hear-and-speak/

2025) partially address the problem but remain *speech-to-text* benchmarks where evaluation ends at the textual response, leaving no benchmark that directly evaluates S2S models' response speech for paralinguistic awareness.

Beyond the lack of benchmarks, no paralinguistic-aware S2S models currently exist. Our study shows that most S2S models fail to appropriately adjust responses according to different speaking styles (e.g., emotional tone) of users, often inferring speaker state from content alone and producing tone-deaf or awkward replies. This limitation stems from existing spoken dialogue datasets, which rarely capture the style dynamics between input and output (Ding et al., 2025; Fang et al., 2025a;b). Collecting such data is expensive, as it requires style annotation and expressive response recording, making data scarcity a critical bottleneck for developing paralinguistic-aware S2S models (Huang et al., 2025b).

Inspired by DeepSeek-R1 (Guo et al., 2025), which incentivizes novel behaviors through RL without SFT demonstrations, we ask *whether paralinguistic-aware dialogue capabilities can similarly emerge via RL with minimal supervision.* To answer, we introduce a novel framework for paralinguistic-aware S2S, **ParaS2S**. ParaS2S comprises a new S2S benchmark **ParaS2SBench** and a RL framework **ParaS2SAlign**. Specifically, **ParaS2SBench** is designed to jointly evaluates both the content and speaking styles of input and output speech. It uniquely covers challenging test queries with controlled scenarios and contrastive speaking styles, properties that enable close examination of models' paralinguistic awareness. We design a data curation pipeline to automatically generate high-quality speech prompts adhering to these properties, covering key paralinguistic aspects—emotion, sarcasm, age, and gender. Using this dataset, we expose the common tone-deaf issue in current S2S models, including state-of-the-art (SOTA) open-source models such as Qwen2.5 Omni (Xu et al., 2025) and Kimi-Audio (Ding et al., 2025), as well as closed-source systems such as ChatGPT Voiced Mode (Achiam et al., 2023).

To enable a faithful proxy for human evaluation, we study automatic judges for **speech-to-speech evaluation**. Recent works have explored leveraging audio large language models (ALLMs) for automatic evaluation Chiang et al. (2025); Zhang et al. (2026). These approaches feed the output audio directly into ALLMs (e.g., ChatGPT) and obtain a judgment score. However, in our study, we find that current ALLMs lack the ability to reliably score the naturalness of a dialogue pair, often hallucinating paralinguistic cues in both the input and output speech based on the spoken content. Instead, we propose to decouple the paralinguistic analysis from the linguistic content with the PolyTone training strategy. In our multi-stage judge, decoupled captioners separately analyzes spoken content and speaking style—including age, gender, emotion, tone—into textual representations, which is then reliably evaluated by large language models (LLMs). By reducing language-based style hallucination, our judge achieves higher correlation with human scoring than ALLM-based judging.

The automatic judge further enables S2S models to explore and learn from unlabeled audio via RL. We verify the effectiveness of the automatic judge in guiding RL post-training with **ParaS2SAlign**. We first distill the pipeline-based judge into a smaller and faster reward model for online learning, and optimize the base S2S model with Group Relative Policy Optimization (GRPO) (Shao et al., 2024). Our results show that while supervised fine-tuning (SFT) is effective and outperforms existing models[3], RL surpasses SFT by more than 10% in response content and speaking-style appropriateness on ParaS2SBench under both objective and subjective evaluations, demonstrating the effectiveness of our automatic judge. Furthermore, in cost-controlled experiments, RL requires only 10 hours of demonstrations as warm-up and achieves the same performance as pure SFT with five times as many demonstrations, highlighting its learning efficiency. Our contributions are multifold:

- We present a novel benchmark, **ParaS2SBench**, for paralinguistic-aware S2S dialogue. It evaluates the naturalness of input–output speech pairs in terms of both content and speaking style at the waveform level. The contrastive speaking-style design further reveals the common tone-deaf issue in S2S models.

- We propose a pipeline-based automatic evaluator for S2S interaction naturalness. Our separate modeling of content and speaking style surpasses end-to-end ALLM-based judging.

- With **ParaS2SAlign**, we show that our judge is accurate enough to guide RL post-training, improving S2S models on ParaS2SBench under both objective and subjective evaluations and achieving state-of-the-art performance.

---

[3]At the cost of requiring expensive and non-scalable demonstrations.

- We demonstrate that RL is more data-efficient than SFT for paralinguistic-aware S2S, reducing the required demonstrations, mitigating the data scarcity and highlighting the importance of scalable automatic evaluation.
- We will open-source data, code, and models to lower the barrier for future research.

## 2 RELATED WORK

### 2.1 SPOKEN DIALOGUE BENCHMARKS

Benchmarks have been proposed to evaluate spoken dialogue models. Table 5 compares key differences across benchmarks. Dynamic-SUPERB (Huang et al., 2024) tests instruction-following on 180 tasks (yu Huang et al., 2025). AudioBench (Wang et al., 2025a) unifies speech/sound understanding and QA. AIR-Bench (Yang et al., 2024) adds speech, sound, music tasks, and a *chat* category. MMAU (Sakshi et al., 2025a) raises difficulty with reasoning-intensive QA. SpokenWOZ (Si et al., 2023) provides large-scale human-to-human dialogue data. VoxEval (Cui et al., 2025) converts MMLU (Hendrycks et al., 2021) to speech to assess model intelligence. VoiceBench (Chen et al., 2024) adds more text-based QA datasets including AlpacaEval (Li et al., 2023), OpenBookQA (Mihaylov et al., 2018), and MMLU-pro (Wang et al., 2024). FullDuplexBench (Lin et al., 2025a) evaluates response timing for full-duplex models. Among these works, ADU-Bench (Gao et al., 2025), SD-eval (Ao et al., 2024), VoxDialogue (Cheng et al., 2025), and StyleTalk (Lin et al., 2024a) evaluate responses under different input speaking styles, but focus only on the dialogue models' output text.[4] In contrast, ParaS2SBench performs end-to-end evaluation on both input and output speech, jointly considering content and speaking style.

### 2.2 SPOKEN DIALOGUE MODELS

**From S2T to S2S dialogue models.** Early Speech-to-Text LLMs equip LLMs with *hearing* capabilities while leveraging textual reasoning for audio interaction (Tang et al., 2024; Hu et al., 2024; Gong et al., 2024). AudioReasoner (Xie et al., 2025) introduces Chain-of-Thought (CoT) reasoning to mitigate hallucination, while Qwen-Audio 1/2 (Chu et al., 2023; 2024) and StepAudio (Huang et al., 2025b) further extend dialogue capabilities to enable spoken agents[5]. Recent works explore Speech-to-Speech LLMs that learn input–output speech interaction end-to-end (Zhang et al., 2023; Défossez et al., 2024). GLM-4-Voice (Zeng et al., 2024) and Step-Audio-AQAA (Huang et al., 2025a) rely on interleaved text and audio tokens for grounded speech generation. LLaMa-Omni (Fang et al., 2025a;b), Freeze-Omni (Wang et al., 2025b) and Mini-Omni (Xie & Wu, 2024) propose fine-tuning techniques to preserve LLM intelligence when adding speech modality. Qwen2.5 Omni (Xu et al., 2025) proposes the thinker-talker architecture, while Kimi-Audio (Ding et al., 2025) introduces a dual-head design for text and audio generation.

**Paralinguistic-aware dialogue models.** Among these models, ParalinGPT (Lin et al., 2024b) and StyleTalk (Lin et al., 2024a) are the first to enable speech-to-text LLMs to respond differently to diverse speaking styles. OmniChat (Cheng et al., 2024) extends the speech-to-text study to multi-turn, paralinguistic-aware dialogues. For speech-to-speech models, GOAT-SLM (Chen et al., 2025) is the only model emphasizing paralinguistic-aware dialogue with a multi-stage SFT pipeline. These works rely on SFT with carefully curated, high-quality data, whereas we explore RL to reduce this reliance.

**RL for dialogue models.** RL has been applied to align spoken dialogue models. Align-SLM (Lin et al., 2025b) follows RLAIF (Lee et al., 2024) and adopts DPO (Rafailov et al., 2023) to improve long-range semantics. Qwen2.5 Omni (Xu et al., 2025) uses WER as a preference signal to ground speech generation. Step-Audio (Huang et al., 2025b) and Step-Audio-AQAA (Huang et al., 2025a) rely on human feedback, which is annotation-heavy. ParaS2SAlign is the first RL framework to model content–style and input-output dynamics using scalable AI feedback.

---

[4]StyleTalk predicts both response text and style in textual format, enabling style learning and evaluation. However, it is limited to the few categorical styles supported by Microsoft Azure TTS, and its format assumption prevents evaluation of S2S models.

[5]The response is usually in text, and the *speaking* capability is enabled by a separate TTS module.

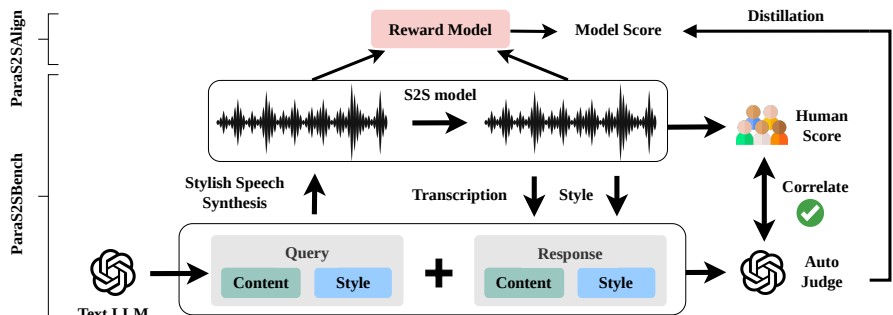

Figure 1: The overall framework of ParaS2S. The bottom part illustrates the dataset construction and automatic judge in ParaS2SBench, while the top part illustrates reward model distillation in ParaS2SAlign. The distilled reward model can then be used by standard RL algorithms such as PPO and GRPO.

## 3    PARAS2SBENCH

ParaS2SBench is uniquely designed to evaluate paralinguistic-aware S2S abilities following three principles:

1. **Contrasting speaking styles.** Following StyleTalk (Lin et al., 2024a), each test query is paired with two *contrasting* speaking styles that demand distinct responses. For example, *"I just bumped into my ex."* may be spoken in either a surprised or sad tone. This design ensures the model to listen to the vocal cues instead of solely relying on the spoken content.

2. **Scenario-controlled queries.** To ensure realistic scenarios, we design each test query to be *paralinguistically relevant* so that speaking style genuinely influences how the response should be generated[6]. To further ensure challenging scenarios, we require the test queries to have *neutral textual content* so that models cannot infer the speaker's state from words alone and must attend to vocal cues[7].

3. **Speech-to-speech evaluation.** Evaluation is performed on output speech, assessing whether the model generates responses with both appropriate content and speaking styles.

In Section 3.1, we describe the curation of the benchmark dataset. In Section 3.2, we present the methodology for automatically evaluating model responses given an input query.

### 3.1    DATASET CONSTRUCTION

In this section, we aim to curate a benchmark dataset, specifically the testing quries, following the first two principles. Please see Table 1 and the demo page[1] for concrete examples. Table 6 shows the statistics. Appendix A.2 provides detailed steps of dataset construction, and we provide the high-level concepts here for conciseness.

**Synthetic speech.**    We design a multi-stage generation and filtering pipeline to generate synthetic test queries. First, an LLM generates scenario-controlled and challenging queries where each specifies one spoken content and two contrasting speaking styles in text. For spoken content, we prompt the LLM the cover a wide range or dialogue topics including interests, work, studies, relationships, travel, health, religion, fashion, finance, and more. For speaking styles, we cover various key paralinguistic factors, including emotion, sarcasm, gender, and age.

---

[6]For example, *"Who is the president of the US?"* is not paralinguistically relevant, since any voicing tone should result in the same response, whereas *"How should I dress for the upcoming wedding?"* is relevant to the speaker's gender.

[7]For example, *"Wow, that's surprising"* clearly expresses surprise and is not textually neutral, yielding a simple test case, whereas *"I just got a call from my boss"* is textually neutral, and the speaker's emotion is difficult to infer without vocal cues.

| Category | Example Dialogue |
|---|---|
| Emotion | **User (happy):** The city is planning to build a new mall near my house. 
 Model (cheerful): That's wonderful! A new mall will bring so many shops and restaurants to explore—sounds exciting! 

 **User (angry):** The city is planning to build a new mall near my house. 
 Model (empathetic): I'm sorry to hear that you're upset about it. Would you like to share what concerns you most about the new mall? |
| Sarcasm | **User (sincere):** It looks like you've outdone yourself with these answers! 
 Model (happy): Thank you! I'm glad my answers were helpful—let me know if you'd like me to expand on anything. 

 **User (sarcastic):** It looks like you've outdone yourself with these answers! 
 Model (apologetic): Hmm, it sounds like you might not be happy with the answers. Could you tell me what was missing or what you'd like done differently? |
| Age | **User (adult):** I need instructions on buying stock with my money. 
 Model (confident): Sure! I can walk you through opening a brokerage account, researching companies, and placing your first order step by step. 

 **User (child):** I need instructions on buying stock with my money. 
 Model (supportive): Investing is exciting! Since you might be under 18, you may need help from a parent or guardian to open a custodial account before you can buy stocks. Would you like me to explain how that works? |
| Gender | **User (male):** Help me pick workout clothes for the gym. 
 Model (professional): Absolutely! Breathable tank tops, moisture-wicking T-shirts, and flexible shorts are great choices. Do you prefer a sporty or casual style? 

 **User (female):** Help me pick workout clothes for the gym. 
 Model (professional): Sure! High-support sports bras, moisture-wicking leggings, and lightweight tops are excellent options. Would you like suggestions for specific brands? |

Table 1: Examples from ParaS2SBench test set. The model responses in gray are for demonstration and not part of the benchmark dataset.

Next, we filter the queries by asking LLM for several checks, including neutrality, reasonability and paralinguistic relevance. Neutrality prevents models from inferring the speaker's state solely from spoken content; reasonability ensures all the content-style pairs are reasonable; paralinguistic relevance ensures that speaking style non-trivially affects the response. If any test is not passed, the query is discard.

Finally, we synthesize spoken queries given the spoken content and style. We leverage the most suitable TTS systems for different styles of speech, and we apply both Emotion2vec- Ma et al. (2024) and WER-based filtering to reject the incorrect generation. Human annotator check is conducted to ensure test set authenticity. We recruit three annotators to manually include only speech prompts with correct content and style from the final filtered set.

**Real speech.**  To further examine model behavior in realistic scenarios, we construct a test set using real speech by filtering queries from existing dialogue datasets. Given the known content and style labels provided by the dataset, we apply filters to check the length[8] and paralinguistic relevance. We rely on two emotion datasets, IEMOCAP (Busso et al., 2008) and MELD (Poria et al., 2019), as they provide sufficient queries that meet our constraints. In contrast, we find it challenging to source enough queries for age and gender from datasets like CommonVoice due to the paralinguistic relevance constraint[9].

## 3.2 AUTOMATIC JUDGE

In ParaS2SBench, subjective evaluation is the gold standard. However, as subjective evaluation is costly, we study how to approximate human judgments with an automatic judge. A straightforward approach is to leverage ALLMs such as ChatGPT or Gemini to directly score naturalness given the user audio and the response audio. However, our preliminary experiments show that these ALLMs

---

[8]In real dialogues, some turns consist of only a few words (e.g., *Haha* or *Sounds good*), which are not suitable for evaluation. We therefore filter out queries with fewer than five words.

[9]For example, in *Could you read the book for me? (female)*, the gender attribute is negligible.

tend to hallucinate paralinguistic properties based on the spoken language, leading to entangled analysis and inaccurate judgments. In contrast, we decouple paralinguistic analysis from linguistic content by using specialized training technique and designing a multi-stage judgment framework.

**Stage 1: PolyTone training.** We train acoustic analysts for emotion, sarcasm, gender, and age by fine-tuning state-of-the-art open-source captioners. To encourage these analysts to attend to vocal cues rather than infer labels from the spoken content, we propose PolyTone training, where the training set consists of utterances spoken in diverse styles but sharing the same linguistic content. This design guides the model to produce different style judgments given identical content and forces it to rely on paralinguistic cues. To curate such parallel data, we use the most suitable TTS systems, as described in Appendix A.2. See Appendix A.3 for model details.

**Stage 2: Separated information extraction.** We extract transcriptions and style labels, including gender, age, emotion, sarcasm, and a natural-language tone description. For transcription, we use Whisper-V3 Radford et al. (2023). For gender, age, emotion, and sarcasm, we leverage the acoustic analysts trained with PolyTone. For the tone descriptor, we use AudioReasoner Xie et al. (2025), which reduces style hallucination via a structured reasoning pattern and is prompted to ignore the spoken content. By using separate, specialized components in our evaluation framework, we avoid hallucinations that are difficult to resolve via prompt engineering and mitigate the black-box nature of end-to-end ALLMs.

**Step 2: LLM analysis on text.** After we explicitly project various aspects of the speech information into text for both input and output speech. A text LLM is then used to analyze a matching score on a 1-5 Likert scale following an expert-designed guideline. We will study its correlation to human scoring, and compare with the ALLM-based judging.

## 4 ParaS2SAlign

To gauge the accuracy and usefulness of the automatic judge in ParaS2SBench, we study its effectiveness in guiding RL post-training of S2S models. Although the pipeline in Section 3.2 is accurate, it is slow in practice: it requires running several style analysts, a reasoning-based speech-to-text LLM, and multiple LLM API calls. This makes typical online RL training impractical when rewards are computed using the automatic-judge pipeline. To address this, we aim to distill the automatic judge pipeline into a single reward model and enable a fast online scoring. We propose a two-stage approach for the distillation. The details are illustrated in Appendix A.4.

**Stage 1: S2S model warmup.** The reward model produces a score given a user speech and a response speech. We aim to collect diverse responses from S2S models given input queries and score them with the automatic judge to serve as training data for the reward model. Unfortunately, existing S2S models do not possess paralinguistic-aware dialogue capabilities, often producing similar responses under contrasting speaking styles. Moreover, it is difficult to obtain satisfactory samples from off-the-shelf S2S models when they behave in a tone-deaf manner, resulting in uniformly low scores. This issue not only hinders reward model learning but also impedes subsequent RL training, as the model cannot learn from positive examples and self-improve. To address this, a warm-up supervised fine-tuning (SFT) stage is critical. We follow similar steps to those in Section 3.1 to construct the training queries. For demonstrations, we use a text LLM to generate suitable response content and style, followed by TTS to synthesize speech and manual selection to ensure expressiveness.

**Stage 2: Reward model distillation.** With the warm-up model, we sample diverse responses for each training query and score naturalness using the full judge pipeline described in Section 3.2. The warm-up model exhibits preliminary paralinguistic-aware dialogue capabilities and begins to respond differently according to the input speaking styles, albeit unstably. As a result, the resulting responses include both high- and low-scoring samples. Using the resulting preference dataset of *(query speech, response speech, score)* triples, we fine-tune Qwen2.5-Omni Xu et al. (2025) with LoRA Hu et al. (2022) to obtain the reward model.

With the reward model ready, we can apply standard RL algorithms to validate the effectiveness of the benchmark judge. We use GRPO Shao et al. (2024) because it is simple and widely adopted. We will show that GRPO significantly improves our S2S model under the judge, as verified by both objective and subjective evaluations. With our pipeline open-sourced, other methods such as PPO Schulman et al. (2017) or DPO Rafailov et al. (2023) can also be used for post-training; we leave this to future work.

# 5 EXPERIMENTS

**Data.** We report benchmark data statistics in Table 6. For PolyTone data, we follow Section 3.1 to create queries with multiple speaking styles. For each category in gender, age, emotion and sarcasm, 10k speech prompts alone with their style lables are generated for fine-tuning captioners. For SFT data, we follow Section 4 to build prompt–demonstration pairs from 10k speech prompts, totaling 100 hours of paired data. We use this SFT data for both the RL warmup (small-scale) setting and the pure SFT (large-scale) setting. Following Section 3.1, we curate up to 100k prompts, each with speech and text annotations for content and style. For reward model distillation, we use up to 10k speech prompts. For each prompt, the warmup model generates 32 completions, yielding 320k prompt–response–score pairs. For RL training, we use all speech prompts; however, we use only the speech and discard all labels during training. For text LLM, We use ChatGPT 4.1 across the paper.

**Training.** For PolyTone training, a single H100 GPU is used. For SFT, we use 8 NVIDIA H100 GPUs with FSDP (Zhao et al., 2023), a learning rate of 1e-5, and a global batch size of 64. For reward model LoRA fine-tuning, we use a single H100 with a learning rate of 1e-6 and a batch size of 10. For RL, we again use 8 H100 GPUs with FSDP, a learning rate of 5e-4, a global query batch size $B$ of 32, and a group size $G$ of 8. Each batch includes 256 scored completions for learning.

**Baselines.** For our multi-stage automatic judge, we ablate the effective of the linguistically decoupled methods such as PolyTone. We use gpt-audio as the end-to-end ALLM baseline.

## 5.1 COMPARING AUTOMATIC JUDGES WITH HUMAN SCORING

We study how to make the automatic judge align with human scoring. For this study, we sampled 200 prompts per paralinguistic category from the benchmark for human annotation. For each speech prompt, we obtain two types of responses:

- **TTS-based responses**: Text LLM generates the response content and style, and diverse TTS systems synthesize speech to simulate different speaking styles. We include YourTTS (Casanova et al., 2022), CosyVoice (Du et al., 2024), Sesame[10], and *gpt-4o-mini-tts*. These systems range from flat and neutral to expressive, spontaneous, and fine-grained controlled styles. We also add a baseline, *gpt-4o-mini-tts (bad)*, where we instruct LLM to produce suboptimal content or style such as tone-deaf content and inappropriate speaking style.
- **S2S model-based responses**: End-to-end S2S models directly generate speech responses. We include GPT-4o Voice mode, Qwen2.5 Omni, and GLM-4-Voice.

TTS-based responses isolate the effect of response tone under identical gold content, while S2S responses reflect real model behavior. Each prompt–response pair is scored by three human experts on a Likert scale[11]. Each input query is then paired with several TTS-based responses and several S2S model responses. For each query–response pair, we have a human final score and several automatic judgment scores from different methods. We compute the Pearson correlation between the judges and human scores in Table 2.

**Multi-stage automatic judges correlate better with human scoring.** In Table 2, our multi-stage judgment framework clearly outperforms end-to-end ALLMs. When provided with ground-truth

---

[10]https://www.sesame.com/research/crossing_the_uncanny_valley_of_voice

[11]We first conducted preliminary annotations to align guidelines and maximize agreement, and discarded official annotations where all three experts disagreed. We use the major vote as the final score.

Table 2: **Correlation comparison.** Correlation between automatic judges and human scores. We ablate different information provided to the text-LLM judge. The transcriptions of both the input and the output are always included. "–" indicates inclusion of ground-truth labels from the dataset; ✗indicates that the information is excluded; ✓indicates inclusion via linguistically decoupled methods (e.g., PolyTone & structural reasoning); ✓indicates inclusion without such methods.

| ID | Input | | | | Output | | Pearson Correlation | | | | |
|----|-----|--------|---------|---------|---------|------|-------|--------|---------|---------|---------|
| | age | gender | emotion | sarcasm | emotion | tone | age | gender | emotion | sarcasm | average |
| **Audio LLMs (baseline)** | | | | | | | | | | | |
| B1 | gpt-audio | | | | | | 0.682 | 0.637 | 0.612 | 0.541 | 0.618 |
| **Multi-stage Automatic Judge (ours)** | | | | | | | | | | | |
| O1 | - | - | - | - | ✗ | ✗ | 0.851 | 0.705 | 0.742 | 0.731 | 0.757 |
| O2 | - | - | - | - | ✗ | ✓ | 0.862 | 0.702 | 0.760 | 0.779 | 0.776 |
| O3 | - | - | - | - | ✓ | ✗ | 0.821 | 0.668 | 0.712 | 0.723 | 0.731 |
| O4 | - | - | - | - | ✓ | ✓ | 0.835 | 0.662 | 0.738 | 0.735 | 0.743 |
| O5 | ✓ | ✓ | ✓ | ✓ | ✗ | ✓ | 0.821 | 0.683 | 0.684 | 0.703 | 0.723 |
| O6 | ✓ | ✓ | ✓ | ✓ | ✗ | ✓ | 0.817 | 0.672 | 0.645 | 0.669 | 0.701 |

paralinguistic labels for the input speech, our pipeline consistently outperforms ALLMs by more than 15% in correlation. When using dedicated captioners to infer the input style, the performance gap remains above 10%. By analyzing the ALLM's reasoning steps, we find that it heavily relies on the spoken content to infer a plausible context for the final judgment, frequently neglecting or hallucinating the relevant paralinguistic cues in contrasting utterances. These results suggest that ALLMs may hallucinate paralinguistic cues and are less stable than the pipeline-based approach.

**Natural-language tone descriptions better capture response style.**   In Table 2 (O1–O4), natural-language tone descriptions (O2) yield substantially higher correlations with the human judgments than emotion-tag conditioning (O3, O4), while emotion-tag conditioning can be comparable to or worse than leaving the output style unspecified (O1). By inspecting the predicted style labels, a plausible explanation is that dialogue *tone* (e.g., *understanding*, *reassuring*, *apologetic*, *professional*) does not map cleanly to discrete *emotion* categories: emotions primarily describe a speaker's affective state Ekman (1992), whereas tone reflects pragmatic/interactional voice functions (e.g., stance or communicative attitude) Vandepitte (1989); Allan et al. (1997). In line with this, an emotion captioner often produces labels that are weakly aligned with the intended attitude[12], which can introduce noise when such labels are used by a text-based LLM judge and thus reduce correlation.

**PolyTone training helps most in capturing true emotion and attitude.**   In Table 2 (O5), replacing the ground-truth labels with model predictions substantially degrades performance, especially in the emotion and sarcasm categories. By contrast, in the age and gender categories, the degradation is minor, as predicting *child/adult* and *male/female* may be easier for state-of-the-art captioners. In Table 2 (O6), the training data construction for the captioners is similar to that of the PolyTone dataset, but the spoken content is always unique. As shown in O5 and O6, the PolyTone training technique yields significant improvements for emotion and sarcasm but smaller gains for age and gender, suggesting the benefit of content-invariant, style-augmented training data.

**Automatic judge preserves the ranking of human scores.**   In Table 3, S2S responses lag significantly behind TTS responses due to the tone-deaf content, where the latter benefit from ground-truth style labels. The scores of S2S responses hover around 3, indicating models fail to adapt to contrasting speaking styles[13]. Second, across all models, the rankings with benchmark and human

---

[12]This issue may be exacerbated because three of Ekman's six basic emotions Ekman (1992) (anger, fear, and disgust) are often undesirable for spoken agents.

[13]For prompts with two contrasting styles, models often score 5 for one response and 1 for the other tone-deaf response, averaging 3.

Table 3: Comparison of automatic judge and human scoring on response ranking. The default automatic judge uses the input ground-truth style labels to avoid error propagation.

| | Age | | Emotion | | Gender | | Sarcasm | | Avg | |
|---|---|---|---|---|---|---|---|---|---|---|
| | Auto | Human | Auto | Human | Auto | Human | Auto | Human | Auto | Human |
| **TTS-based** | | | | | | | | | | |
| gpt-4o-mini-tts (good) | 4.420 | 4.380 | 4.646 | 4.654 | 4.739 | 4.506 | 4.790 | 4.337 | 4.649 **(1)** | 4.469 **(1)** |
| gpt-4o-mini-tts (bad) | 1.215 | 1.177 | 1.159 | 1.041 | 1.325 | 1.590 | 1.210 | 1.251 | 1.227 **(8)** | 1.265 **(8)** |
| Sesame | 4.412 | 4.216 | 4.512 | 4.324 | 4.701 | 4.332 | 4.71 | 4.182 | 4.583 **(2)** | 4.263 **(2)** |
| CosyVoice | 4.380 | 3.994 | 4.417 | 4.012 | 4.612 | 4.201 | 4.680 | 3.864 | 4.522 **(3)** | 4.018 **(3)** |
| YourTTS | 4.410 | 4.037 | 4.302 | 3.801 | 4.534 | 4.230 | 4.580 | 3.804 | 4.457 **(4)** | 3.968 **(4)** |
| **S2S models** | | | | | | | | | | |
| gpt-4o-audio-preview | 2.685 | 2.630 | 3.711 | 2.713 | 3.096 | 3.682 | 2.815 | 2.611 | 3.077 **(6)** | 2.909 **(5)** |
| Qwen2.5 Omni | 2.930 | 2.728 | 3.680 | 2.522 | 2.933 | 3.493 | 2.910 | 2.509 | 3.113 **(5)** | 2.863 **(6)** |
| GLM-4-Voice | 2.570 | 2.493 | 3.489 | 2.384 | 2.821 | 3.521 | 2.720 | 2.301 | 2.900 **(7)** | 2.675 **(7)** |

Table 4: Comparing paralinguistic-aware dialogue capabilities with ParaS2SBench score.

| | Synthetic | | | | | Real | | | Avg |
|---|---|---|---|---|---|---|---|---|---|
| | Age | Emotion | Gender | Sarcasm | Avg | IEMOCAP | MELD | Avg | |
| ***Baseline*** | | | | | | | | | |
| Whisper-GPT-TTS | 3.050 | 3.121 | 2.916 | 3.005 | 3.022 | 3.562 | 3.412 | 3.487 | 3.176 |
| ***Closed Source*** | | | | | | | | | |
| gpt-4o-audio-preview | 3.205 | 3.633 | 3.342 | 2.957 | 3.284 | 3.770 | 3.508 | 3.639 | 3.403 |
| Gemini | 3.301 | 3.811 | 3.413 | 3.263 | 3.447 | 3.813 | 3.712 | 3.762 | 3.552 |
| ***Open Source*** | | | | | | | | | |
| Qwen2.5 Omni | 3.170 | 3.653 | 3.236 | 2.935 | 3.248 | 3.626 | 3.599 | 3.612 | 3.369 |
| GLM 4 | 2.885 | 3.447 | 2.976 | 2.803 | 3.033 | 2.934 | 3.141 | 3.037 | 3.034 |
| LLaMa-Omni 2 | 3.123 | 3.512 | 3.064 | 3.164 | 3.215 | 3.425 | 3.462 | 3.443 | 3.291 |
| Freeze-Omni | 2.819 | 2.316 | 2.884 | 2.701 | 2.680 | 2.835 | 3.061 | 2.948 | 2.769 |
| Kimi-Audio | 3.141 | 2.673 | 3.091 | 2.665 | 2.892 | 1.365 | 1.166 | 1.265 | 2.350 |
| ***Ours*** | | | | | | | | | |
| Kimi-Audio SFT | 4.393 | 4.090 | 3.530 | 4.291 | 4.076 | 4.121 | 3.307 | 3.714 | 3.955 |
| Kimi-Audio GRPO | 4.496 | 4.490 | 4.239 | 4.538 | 4.441 | 4.394 | 3.927 | 4.161 | 4.382 |
| ***Topline*** | | | | | | | | | |
| GPT-TTS | 4.525 | 4.691 | 4.812 | 4.791 | 4.705 | 4.710 | 4.824 | 4.766 | 4.725 |

scores are nearly identical, with only one switch. The rankings of TTS systems are also consistent: *gpt-4o-mini-tts* > Sesame > CosyVoice > YourTTS[14].

## 5.2 VERIFYING THE EFFECTIVENESS OF AUTOMATIC JUDGE FOR RL

**Settings.** We study the effectiveness of the automatic judge to guide RL post-training. Note that in Table 2, O2 stands for the highest correlation with the ground-truth style labels. We use this pipeline as the default *benchmarking judge* to serve as a proxy for the human evaluation. Note that we also conduct human evaluation in Appendix A.6. On the other hand, for the reward model distillation in Section 4, we use O5 which enables the pipeline to judge unlabeled input-output pairs, a more realistic scenario. We use Kimi-Audio Ding et al. (2025) as the base S2S model as is possess the highest intelligence among the open-source S2S models.

In Table 4, the Whisper-GPT-TTS pipeline uses Whisper-v3 to transcribe the input query without considering the speaking style, generates the response text with LLM, and synthesizes speech with *gpt-4o-mini-tts*. This pipeline serves as a baseline where speaking style is ignored. The topline, on the other hand, leverages the ground-truth transcription and style label of the query to generate both the response content and style with LLM, and then synthesizes expressive speech using *gpt-4o-mini-tts*.

---

[14]Since the four TTS systems share the same response content, their scoring differences stem from the speaking style. We observe that AudioReasoner tends to classify CosyVoice and YourTTS outputs as calm, neutral, or flat, which is less empathetic.

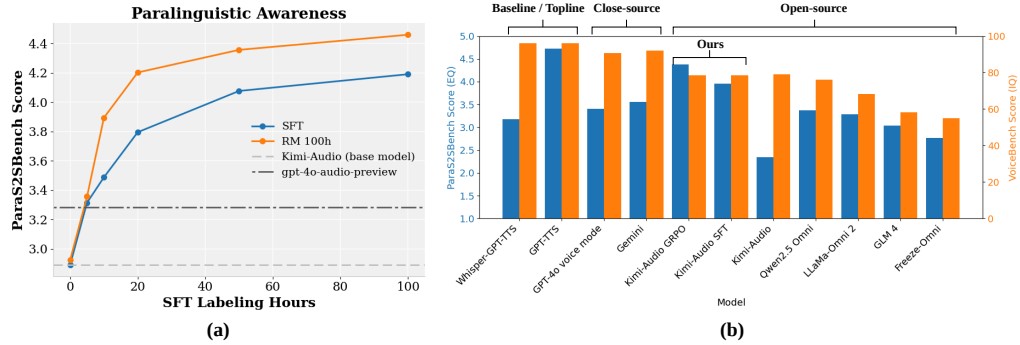

Figure 2: (a) Studying the effectiveness of RL across different labeled data regimes. (b) Comparing the paralinguistic-aware dialogue capabilities and original capabilities across models.

**Automatic Judge help guiding RL post-training.**   In Table 4, GRPO clearly helps improve the performance upon SFT on the paralinguistic-aware dialogue, with over 10% overall improvement, including both synthetic and real speech.

**Existing S2S models fail to adjust responses upon different speaking styles.**   Table 4 shows that most existing S2S models perform similarly to the pipeline baseline, suggesting that they do not account for the input speaking style and produce similar responses even for contrasting queries. In contrast, our models achieve more than a 68% improvement over the base model and surpasses all existing models. Human evaluation in Appendix A.6 further corroborates the findings.

**RL with a judge helps mitigate SFT data scarcity.**   As shown in Figure 2(a), RL continues to benefit the SFT model across most data regimes, from 10 to 100 hours of SFT data, suggesting the effectiveness of the automatic judge and RL post-training.[15] Notably, Figure 2 shows that the post-trained model with only 20 hours of warmup data surpasses the SFT model trained with the full 100 hours of SFT data. This suggests that RL mitigates data scarcity for paralinguistic-aware dialogue and may be a more suitable approach for this problem. These results further highlight the importance of a scalable and accurate automatic judge for speech-to-speech interaction, which serves as a key component of RL.

**The original capabilities of the base model are preserved.**   Finally, we benchmark the original capabilities of the warmup and post-trained models on VoiceBench Chen et al. (2024). Figure 2 shows that these original capabilities are largely preserved. We attribute this to a lightweight SFT warmup (only 2 epochs), along with KL-regularized GRPO to prevent the policy from drifting too far from the base model (Appendix A.7). In contrast, when we experimented with pure SFT for more than 10 epochs, we observed degradation in the original capabilities, especially in response length: the model starts to favor short responses resembling those in our SFT data even for queries that ask for detailed information, suggesting overfitting.

## 6    CONCLUSION

We present ParaS2S, a framework designed for paralinguistic-aware speech-to-speech interaction. We formulate the problem and construct a benchmark dataset covering diverse scenarios and multiple paralinguistic aspects, including both synthetic and real speech. We provide an automatic judge that correlates well with human preferences to enable model scoring. We demonstrate the effectiveness and efficiency of exploring on unlabeled speech and learning from the judge's signal. With our judge and GRPO, we enable state-of-the-art paralinguistic-aware dialogue capabilities using only 10 hours of warm-up demonstrations, consistently demonstrating superior label efficiency compared to pure SFT. We will release the data, models, and code to lower the barrier for future research.

---

[15]With only five hours of SFT data, the warmup model cannot sample sufficiently good responses to learn a reliable reward model, resulting in no improvement over SFT after post-training.

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

# A APPENDIX

## A.1 RELATION TO PRIOR S2T & S2S BENCHMARKS

We compare our ParaS2SBench to existing S2T & S2S Benchmarks in Table 5. Note that ParaS2SBench is the first benchmark to directly assess the speaking style of the response speech at the waveform level.

| Benchmarks | Task type | | Evaluate Input | | Evaluate Output | | Style Dimension | |
|---|---|---|---|---|---|---|---|---|
| | Und. | Dia. | Content | Style | Content | Style | Para. | Speaker |
| *Speech-to-Text Evaluation* | | | | | | | | |
| Dynamic-SUPERB | ✓ | ✗ | ✓ | ✓ | ✓ | ✗ | ✓ | ✓ |
| AudioBench | ✓ | ✗ | ✓ | ✓ | ✓ | ✗ | ✓ | ✓ |
| AIR-Bench | ✓ | ✓ | ✓ | ✓ | ✓ | ✗ | ✓ | ✓ |
| MMAU | ✓ | ✗ | ✓ | ✓ | ✓ | ✗ | ✓ | ✓ |
| VoiceBench | ✓ | ✓ | ✓ | ✗ | ✓ | ✗ | ✗ | ✗ |
| ADU-Bench | ✓ | ✓ | ✓ | ✓ | ✓ | ✗ | ✓ | ✗ |
| SD-Eval | ✗ | ✓ | ✓ | ✓ | ✓ | ✗ | ✓ | ✓ |
| VoxDialogue | ✗ | ✓ | ✓ | ✓ | ✓ | ✗ | ✓ | ✓ |
| StyleTalk | ✗ | ✓ | ✓ | ✓ | ✓ | ✓ | ✓ | ✗ |
| *Speech-to-Speech Evaluation* | | | | | | | | |
| VoxEval | ✓ | ✗ | ✓ | ✓ | ✓ | ✗ | ✓ | ✓ |
| ParaS2SBench (Ours) | ✗ | ✓ | ✓ | ✓ | ✓ | ✓ | ✓ | ✓ |

Table 5: Comparison of spoken dialogue benchmarks. Und. stands for Understanding; Dia. stands for Dialogue; Para. stands for Paralinguistic information.

## A.2 STATISTICS AND DETAILS OF DATASET CONSTRUCTION

Table 6: Statistics of the ParaS2SBench test set.

| | # Prompts | # Utterance | Avg Duration | Hours | Labels |
|---|---|---|---|---|---|
| *Synthetic Speech* | | | | | |
| Emotion | 300 | 600 | 4.59 | 0.77 | Happy, Surprised, Sad, Angry, Fear, Disgust |
| Sarcasm | 300 | 600 | 6.23 | 1.04 | Sincere, Sarcastic |
| Age | 300 | 600 | 4.72 | 0.79 | Adult, Child |
| Gender | 300 | 600 | 4.48 | 0.74 | Male, Female |
| *Real Speech* | | | | | |
| IEMOCAP | 709 | 709 | 10.21 | 2.01 | Happy, Surprised, Sad, Angry, Fear, Disgust |
| MELD | 781 | 781 | 11.31 | 2.45 | Happy, Surprised, Sad, Angry, Fear, Disgust |
| Total | 2690 | 3890 | 6.92 | 7.8 | |

We present the benchmark dataset statistics in Table 6. We detail the steps involved in constructing the benchmark dataset. Note that the training speech prompts for SFT, the reward model, and RL follow the same construction pipeline; however, the training data does not undergo human verification due to the high cost.

1. **Candidate Generation.** We first generate a large corpus of queries with LLM, each consisting of a input spoken sentence $c_i \in \Sigma^*$ followed by two *contrasting* speaking styles, $s_i^1, s_i^2 \in \Sigma^*$, that demand different responses. In the generation prompt, we instruct LLM to

cover diverse topics and scenarios, including interests, work, studies, relationships, travel, health, religion, fashion, finance, and more.[16]

2. **Script Quality Filtering.** For each spoken content $c_i$, we construct two queries, $(c_i, s_i^1)$ and $(c_i, s_i^2)$. For each query $(c_i, s_i)$, we control the scenario by asking LLM for several checks, including neutrality, reasonability and paralinguistic relevance. Neutrality prevents models from inferring the speaker's state solely from text $c_i$; reasonability ensures the content $c_i$ and style $s_i$ is a reasonable pair; paralinguistic relevance ensures that speaking style non-trivially affects the response. If any test is not passed, the query is discard. In the following, we provide the detailed illustration for the quality tests:

   (a) **Paralinguistic Relevance Test.** To ensure that the speaking style is non-trivial to the dialogue scenario and meaningfully affects the response, we test whether different speaking styles lead to different responses. We ask an LLM to generate two responses—both content and style—based on the same input content but different speaking styles, once for each style. We then use the LLM to determine whether the two responses exhibit a significant difference. If the responses are similar, implying that the speaking style does not meaningfully affect the outcome, we discard the test case. This test is essential because many spoken queries and their appropriate responses are irrelevant to how they are voiced, such as *"Who is the president of the US?"*.

   (b) **Reasonability Test.** Due to hallucinations, the LLM sometimes generates queries in which the content and speaking style are mismatched. For example, *I want to get screened for cervical cancer. (male/female)* is reasonable for a female speaker but sounds odd for a male speaker. We ask the LLM to evaluate the reasonability of both speaking styles and discard queries that contain one or more unreasonable cases. After the first two tests, we obtain realistic test cases for evaluating paralinguistic-aware dialogue capabilities.

   (c) **Neutrality Test.** We frequently observe that S2S models respond empathetically by inferring from the spoken content rather than relying on paralinguistic cues in the speech signal. For example, *Wow! That's big news!* is almost always associated with a surprised emotion, and *Oh... I got my period* is most likely to be spoken by a female speaker in a sad tone. To make our test cases more challenging and to examine whether S2S models truly attend to the audio, we design test cases using paralinguistically neutral content—utterances for which it is difficult to infer emotion, attitude, gender, or age from text alone. In this way, the model must rely on the audio signal to respond appropriately. In practice, for each query, we ask the LLM whether the spoken sentence is more likely to be voiced in one speaking style, another speaking style, or whether it is neutral and difficult to determine. We then discard queries for which the answer is not neutral.

3. **Speech Synthesis.** We synthesize input waveform $w_i \in R^*$ given the $(c_i, s_i)$ pair. For emotion and sarcasm, we rely on the instruction-based TTS system *gpt-4o-mini-tts*[17]. This system requires a style description, which we generate with LLM based on the style label $s_i$. Since *gpt-4o-mini-tts* supports only a limited number of speakers, we use CosyVoice (Du et al., 2024) for in-context zero-shot synthesis of gender and age. The voice samples for gender are drawn from LibriSpeech (Panayotov et al., 2015) and CommonVoice (Ardila et al., 2020), while the samples for age are drawn from NNCES[18]. We discard samples whose WER with the ground truth exceeds a threshold. For emotion, we further discard samples whose Emotion2vec (Ma et al., 2024) classifier scores are too low (Cheng et al., 2025).

4. **Train/Test Split.** To avoid overlap between training and testing, we use disjoint query topics and TTS speakers.

5. **Human Check.** To ensure test set authenticity, we recruit three annotators to manually include only speech prompts with correct content and style from the filtered set.

---

[16]The prompt also includes explicit instructions for our quality criteria. However, we find that LLM sometimes hallucinates and struggles to follow instructions and generate high-quality queries in a single pass, making subsequent quality filtering necessary.

[17]https://www.openai.fm/

[18]https://www.kaggle.com/datasets/kodaliradha20phd7093/nonnative-children-english-speech-nnces-corpus . We do not use MyST (Pradhan et al., 2024) since the data link is unavailable.

Finally, we construct a testing set $D_{\text{test}} = \{(c_i, s_i, w_i)\}$ where $c_i \in \Sigma^*$ is the input spoken content, $s_i \in \Sigma^*$ is the input speaking style, and $w_i \in R^*$ is the input audio prompt.

### A.3 DETAILS OF AUTOMATIC JUDGE

Given an input query $(c_i, s_i, w_i) \sim D_{\text{test}}$, the S2S model $M$ samples a response speech $w_o \sim \pi_M(O|w_i)$. We project both the content and the speaking style of $w_i$ and $w^o$ into text. We denote $c_i$ and $s_i$ as the content and style of the input; $c_o$ and $s_o$ as the content and style of the output.

For content, Whisper-V3 (Radford et al., 2023) is used to get the transcription. The style description is composed of emotion, sarcasm, gender, age and tone descriptions. We fine-tune state-of-the-art captioners for different paralinguistic properties with the PolyTone strategy. We fine-tune Emotion2vec Ma et al. (2024) for emotion and sarcasm classifiers. We use a wav2vec 2.0-based age classifier[19] and a ECAPA-TDNN-based gender classifier[20] on Huggingface. We leverage AudioReasoner (Xie et al., 2025) to extract output speaking tone. AudioReasoner equips Qwen-Audio 2 (Chu et al., 2024) with reasoning capabilities by distilling Chain-of-Thought (CoT) paths from Gemini (Team et al., 2024) to reduce hallucination.

Finally, given the input content $c_i$ and style $s_i$, along with the extracted output content $c_o$ and style $s_o$, we use LLM to score the naturalness following the guideline $r$ designed by human experts, described in the Appendix A.8.

$$f_{\text{auto}} = LLM(c_i, s_i, c_o, s_o, r) \tag{1}$$

We show that this scoring pipeline can align with human judgments $f_{expert}$ in Section 5.1. Both $f_{auto}$ and $f_{expert}$ are on a 1–5 Likert scale.

### A.4 DETAILED FORMULATION OF PARAS2SALIGN

We design a three-stage online RL framework that uses a reward model to approximate the automatic judge and employs GRPO (Shao et al., 2024). We use Kimi-Audio (Ding et al., 2025) as the base model $\theta_{\text{base}}$[21], while the framework can be applied to any LM-based S2S model. For Kimi-Audio, the audio input $w_i$, text input $c_i$, audio output $w_o$, and text output $c_o$ are preprocessed and organized into four token streams: $a_i, t_i \in \mathbb{Z}^{L_i}$ and $a_o, t_o \in \mathbb{Z}^{L_o}$. The input streams $(a_i, t_i)$ are padded to the same length $L_i \in \mathbb{Z}$, and the output streams to $L_o \in \mathbb{Z}$. The input embeddings of the audio and text streams are summed before being fed into the Transformer, and from the middle of the model, two prediction heads predict the next token for each stream.

$$\pi_\theta(a_o, t_o \mid a_i, t_i) = \prod_{n=1}^{|a_o|} \pi_\theta(a_{o,n}, t_{o,n} \mid a_{o,<n}, t_{o,<n}, a_i, t_i) \tag{2}$$

For inference, output audio and text tokens are sampled auto-regressively $(a_o, t_o) \sim \pi_\theta(O \mid a_i, t_i)$. Audio tokens are decoded into the sampled waveform with a flow-matching decoder: $w_o = \rho(a_o)$.

**Stage 1. Warm-up.** SFT serves as a crucial warm-up stage for RL, as we observe that existing S2S models lack paralinguistic-aware dialogue capabilities. Consequently, they fail to sample high-quality responses and cannot provide a useful learning signal for RL. To construct the SFT dataset $D_{\text{sft}}$, we follow Section 3.1 to generate a training set of speech queries with both input content $c_i$ and style labels $s_i$. For each query $(c_i, s_i)$, we use LLM to produce the most suitable response $(c_o, s_o)$, including both a textual transcription and a tone description. We then synthesize the expressive response $w_o$ using *gpt-4o-mini-tts*. Because *gpt-4o-mini-tts* can be unstable, we synthesize 10 candidates, apply WER-based filtering, and perform manual selection to obtain high-quality warm-up

---

[19]https://huggingface.co/audeering/wav2vec2-large-robust-24-ft-age-gender

[20]https://huggingface.co/JaesungHuh/voice-gender-classifier

[21]Since it exhibits high intelligence and strong dialogue capabilities (Chen et al., 2024) and is fully open-sourced. We do not use Qwen2.5-Omni (Xu et al., 2025) because its speech tokenizer is not released, making S2S fine-tuning infeasible.

demonstrations $w_o$. With the input–output mapping $D_{\text{sft}} = \{(w_i, w_o, c_i, c_o)\}$, we train next-token prediction on both the preprocessed audio stream $a_i \| a_o$ and the text stream $t_i \| t_o$ by optimizing $\theta$ for higher likelihood $\mathbb{E}_{D_{\text{sft}}}[\pi_\theta(a_o, t_o \mid a_i, t_i)]$, initializing from $\theta_{\text{base}}$ and obtaining $\theta_{\text{sft}}$.

**Stage 2. Distilling Reward Model.** To distill our benchmark pipeline into a reward model, we construct a preference dataset $D_{\text{prefer}}$. We first prepare $Q$ speech queries $\{(c_i^j, s_i^j, w_i^j)\}_{j=1}^Q$ following Section 3.1. The SFT model now possesses preliminary paralinguistic-aware dialogue capabilities and begins to respond differently according to the input speaking styles, but unstably. Each query $(c_i, s_i, w_i)$ is preprocessed into input token streams $(a_i, t_i)$. We sample $K$ diverse speech responses with high sampling temperature, $(a_o, t_o) \sim \pi_\theta(O \mid a_i, t_i), w_o = \rho(a_o)$. We then score the resulting $Q \times K$ query–response pairs following Equation 1 to construct a preference dataset $D_{\text{prefer}} = \{(w_i, w_o, f_{\text{gpt}})\}$, where $f_{\text{gpt}}$ is the fitness score of $w_i$ and $w_o$, depending on content label $c_i$, style label $s_i$, extracted content $c_o = C(w_o)$ and extracted style $s_o = S(w_o)$. Finally, we use LoRA (Hu et al., 2022) to fine-tune Qwen2.5-Omni (Xu et al., 2025) as the reward model, which is employed as a Speech-to-Text LLM. The model takes the query speech, response speech, and scoring guideline $r$ as input, and outputs a single score on a Likert scale. We denote the reward model as $\phi$. The score is treated as a single character and optimized with the cross entropy loss: $\mathbb{E}_{D_{\text{prefer}}} \phi(f_{\text{gpt}} \mid w_i, w_o, r)$.

**Stage 3. Post-training.** Using the warm-up model $\theta_{\text{sft}}$ and the reward model $\phi$, we enable the model to explore the search space for higher scores via GRPO (Shao et al., 2024) on the large set of unlabeled speech. We do not use PPO (Schulman et al., 2017) due to its substantial memory and computational burden of the value function. Moreover, in our case, only the last token of the response is assigned a final reward, which complicates the training of the value function that needs to be accurate at every token (Shao et al., 2024). Given the unlabeled speech prompt dataset $D_{\text{rl}} = \{w_i\}$, we obtain the transcription with Whisper-v3 and construct input token streams $D'_{\text{rl}} = \{(w_i, a_i, t_i)\}$. We optimize $\theta_{\text{sft}}$ to maximize the objective:

$$\mathcal{J}_{\text{GRPO}}(\theta) = \mathbb{E}\big[(w_i, a_i, t_i) \sim D'_{\text{rl}}, \{(a_o^g, t_o^g)\}_{g=1}^G \sim \pi_{\theta_{\text{old}}}(O \mid a_i, t_i)\big]$$

$$\frac{1}{G} \sum_{g=1}^G \frac{1}{|a_o^g|} \sum_{n=1}^{|a_o^g|} \left\{ \min\left[ \frac{\pi_\theta(a_{o,n}^g, t_{o,n}^g \mid a_i, t_i, a_{o,<n}^g, t_{o,<n}^g)}{\pi_{\theta_{\text{old}}}(a_{o,n}^g, t_{o,n}^g \mid a_i, t_i, a_{o,<n}^g, t_{o,<n}^g)} \hat{A}^g, \right.\right.$$

$$\left.\left. \text{clip}\left( \frac{\pi_\theta(a_{o,n}^g, t_{o,n}^g \mid a_i, t_i, a_{o,<n}^g, t_{o,<n}^g)}{\pi_{\theta_{\text{old}}}(a_{o,n}^g, t_{o,n}^g \mid a_i, t_i, a_{o,<n}^g, t_{o,<n}^g)}, 1-\epsilon, 1+\epsilon \right) \hat{A}^g \right] - \beta \, \mathbb{D}_{\text{KL}}[\pi_\theta \| \pi_{\text{ref}}] \right\}, \tag{3}$$

$$\mathbb{D}_{KL}[\pi_\theta \| \pi_{ref}] = \frac{\pi_{\text{ref}}(a_{o,n}^g, t_{o,n}^g \mid a^i, t^i, a_{o,<n}^g, t_{o,<n}^g)}{\pi_\theta(a_{o,n}^g, t_{o,n}^g \mid a^i, t^i, a_{o,<n}^g, t_{o,<n}^g)}$$

$$- \log \frac{\pi_{\text{ref}}(a_{o,n}^g, t_{o,n}^g \mid a^i, t^i, a_{o,<n}^g, t_{o,<n}^g)}{\pi_\theta(a_{o,n}^g, t_{o,n}^g \mid a^i, t^i, a_{o,<n}^g, t_{o,<n}^g)} - 1, \tag{4}$$

We sample prompts from $D'_{\text{rl}}$, generate $G$ responses, decode tokens into waveforms with $\rho$, score them with $\phi$, compute the normalized advantage $\hat{A}^g = (\phi(w_i, \rho(a_o^g), r) - \mu)/\sigma$, and update the policy $\theta$ for higher rewards. $\mu$ and $\sigma$ are the mean and standard deviation of the raw scores within a group. $\epsilon$ is the clipping threshold.

## A.5 ABLATION FOR GRPO TRAINING

**Global batch size.** The global query batch size defines the total number of distinct speech prompts across devices. Figure 3(a) shows that the ParaS2SBench score continues to improve with larger global batch sizes, while exhibiting diminishing returns as the computing requirement (more GPUs) increases. We use a batch size of 32 as the default, where 8 NVIDIA H100s are sufficient for a single run.

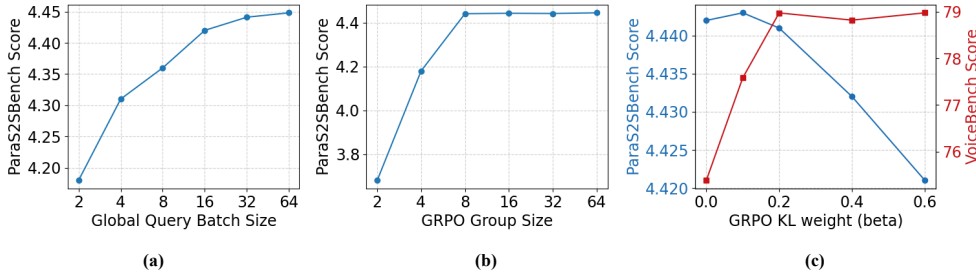

Figure 3: Ablating the effect of global batch size and GRPO's group size and KL penalty weight. For all experiments, we optimize for the same number of steps.

**GRPO group size.** GRPO group size defines how many samples are drawn for each speech prompt. Since GRPO relies on differences between samples for the learning signal, it is crucial to have a large enough group size to ensure diversity. Figure 3(b) shows that when the group size is smaller than 8, performance drops significantly. For example, when the group size is 2, the two samples often receive the same score, providing no learning signal. Interestingly, we find that a group size of 8 is sufficient for effective learning, and increasing the group size further does not provide additional gains.

**KL weight is critical for preserving original capabilities.** Finally, we study the effect of the KL penalty weight $\beta$. During GRPO, we aim to enable paralinguistic-aware dialogue capabilities without degrading the original dialogue capabilities, as training might otherwise overfit to the training set. We leverage VoiceBench (Chen et al., 2024) to quantify changes in the original dialogue capabilities. The benchmark includes daily QA, knowledge-intensive QA, instruction-following tasks in both close-ended and open-ended scenarios. Higher VoiceBench scores indicate stronger general dialogue capabilities, while higher ParaS2SBench scores indicate stronger paralinguistic-aware dialogue capabilities. Figure 3(c) shows that: (1) without a KL penalty, the model suffers from catastrophic forgetting and VoiceBench performance drops significantly; (2) with too high a KL penalty, the model is overly constrained by the original parameters and cannot freely explore the search space, leading to a drop in ParaS2SBench score. We therefore set the default to $\beta = 0.2$, which achieves both capabilities without one degrading severely.

## A.6 HUMAN EVALUATION

Table 7: Comparing paralinguistic-aware dialogue capabilities with human evaluation.

| | Synthetic | | | | | Real | | | Avg |
|---|---|---|---|---|---|---|---|---|---|
| | Age | Emotion | Gender | Sarcasm | Avg | IEMOCAP | MELD | Avg | |
| ***Baseline*** | | | | | | | | | |
| Whisper-GPT-TTS | 3.212 | 3.041 | 3.042 | 3.112 | 3.102 | 3.601 | 3.552 | 3.487 | 3.230 |
| ***Closed Source*** | | | | | | | | | |
| GPT-4o voice mode | 3.375 | 3.833 | 3.542 | 3.078 | 3.457 | 3.862 | 3.694 | 3.778 | 3.564 |
| ***Open Source*** | | | | | | | | | |
| Qwen2.5 Omni | 3.352 | 3.953 | 3.496 | 3.131 | 3.483 | 3.713 | 3.581 | 3.647 | 3.538 |
| GLM 4 | 3.012 | 3.514 | 3.228 | 2.781 | 3.134 | 3.521 | 3.325 | 3.423 | 3.230 |
| Kimi-Audio | 3.278 | 2.382 | 3.121 | 2.912 | 2.924 | 2.231 | 2.272 | 2.252 | 2.699 |
| ***Ours*** | | | | | | | | | |
| Kimi-Audio SFT | 4.192 | 4.223 | 3.812 | 4.131 | 4.089 | 4.212 | 3.407 | 3.810 | 3.996 |
| Kimi-Audio GRPO | 4.316 | 4.510 | 4.381 | 4.422 | 4.407 | 4.336 | 3.859 | 4.098 | 4.303 |
| ***Topline*** | | | | | | | | | |
| GPT-TTS | 4.752 | 4.889 | 4.923 | 4.813 | 4.844 | 4.911 | 4.925 | 4.918 | 4.922 |

In the main article, we present the objective evaluation using the automatic ParaS2SBench score. Although the ParaS2SBench score shows a high correlation with human judgments in Section 5.1,

the correlation remains below 0.9, leaving room for inconsistencies. We therefore study the effectiveness of our approach under human subjective evaluation. Specifically, we crowd-source 10 participants outside our expert annotation group, which designed the scoring guideline $r$ and annotated the preference scores in Section 5.1. These participants have minimal knowledge of the project, including the guideline $r$, to avoid inductive bias. They are given pairs of input and response audio clips and asked to assign a 1–5 mean opinion score based on how naturally the two clips fit together in dialogue. Due to annotation costs, we sample a subset from the ParaS2SBench test set, with 30 prompts per category. For each prompt–response pair, 10 human scores are collected and averaged as the final score.

**Consistency between human evaluation and automatic judgment.** Table 7 shows that the overall trend is consistent with Table 4. SFT on Kimi-Audio provides a significant boost over the base model and surpasses existing models. Kimi-Audio with GRPO further outperforms SFT by 7.6%. Similar to Table 4, the existing models struggle on paralinguistic-aware dialogue.

**Differences between crowd-sourced and expert annotators.** One notable difference between the objective and subjective evaluations is that our crowd-sourced participants tend to assign higher scores than both the benchmark pipeline and our expert annotators. This may be because the participants are not trained to recognize fine-grained paralinguistic labels in speech[22] and often assign high scores when stylistic deviations are subtle[23]. This suggests that in everyday use, typical users may be more tolerant of paralinguistic unawareness or tone-deaf responses than our benchmark criteria imply, which explains the smaller relative improvement compared to the objective evaluation. Nevertheless, the 7.6% relative improvement in the subjective evaluation remains substantial, underscoring the importance of paralinguistic awareness for improving user satisfaction.

## A.7 ANALYSIS ON THE ORIGINAL CAPABILITIES PRESERVATION

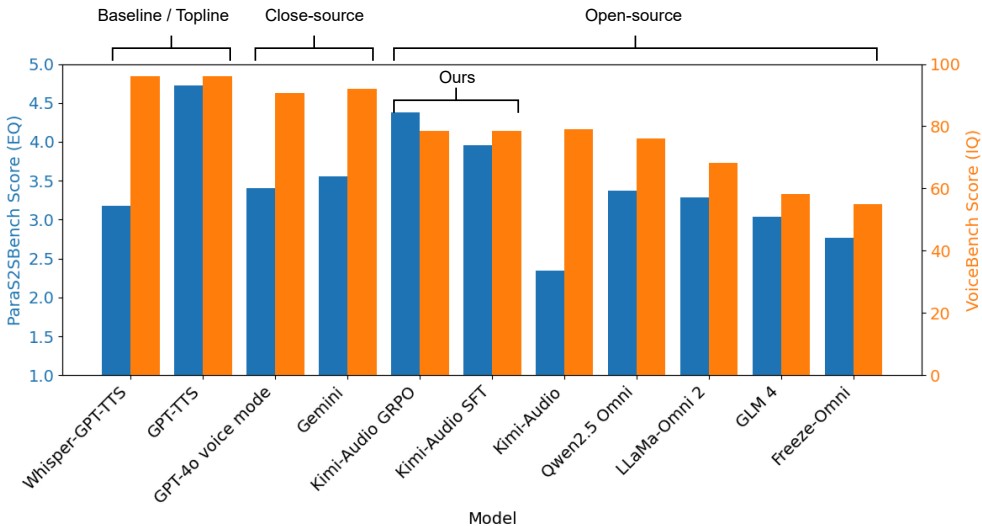

Figure 4: Comparing the overall intelligence and paralinguistic-aware dialogue capabilities across models.

As discussed in Appendix A.5, we maintain the base model's intelligence by carefully tuning the KL penalty. We leverage VoiceBench (Chen et al., 2024) to quantify changes in general intelligence. The benchmark includes daily QA, knowledge-intensive QA, and instruction-following tasks in both

---

[22]They are only instructed to pay attention to speaking style, age, and gender, but are not given detailed style labels to avoid inductive bias.

[23]For example, a slightly sad expression may be perceived as neutral, and an otherwise normal response may still receive a high score.

close-ended and open-ended scenarios. Higher VoiceBench scores indicate stronger general intelligence, while higher ParaS2SBench scores indicate better paralinguistic-aware dialogue capabilities.

**Our post-trained model preserves its top-tier intelligence.** Figure 4 shows that the pipeline-based baseline and topline achieve the highest intelligence, followed by closed-source models. Among all open-source models, ours demonstrates the highest intelligence. This stems from our choice of Kimi-Audio as the base model—since it already possesses the strongest intelligence among existing open-source models—and from our training procedure, which preserves its original capabilities.

**State-of-the-art paralinguistic-aware dialogue capabilities without sacrificing intelligence through RL.** For paralinguistic-aware dialogue capabilities, our SFT model surpasses all existing models except the topline, which leverages ground-truth style labels. Our GRPO model further improves upon the SFT model without compromising intelligence. Overall, our models achieve state-of-the-art performance in both general intelligence and paralinguistic-aware dialogue capabilities.

## A.8 PROMPTS

We include all the prompts used for data generation and automatic scoring.

### A.8.1 QUERY GENERATION FOR AGE

> **Prompt Template:**
> Give me some kids' requests that should not be directly processed due to adult-level permission or qualification. Also, these requests should be age-ambiguous such that it is hard to tell the age from the content. The request should be in oral form suitable for text-to-speech. I want the specific user chats instead of the category or topic names. Give me a DIVERSE dataset of 100 requests related to {*topic*} in the following format:
>
> 1. user_chat1
> 2. user_chat2
> 3. ...
>
> You should give ALL the requests without ending early.
>
> **Topics:**
> - Location & Travel
> - Privacy & Security
> - Interpersonal & Social
> - Mental Health & Emotions
> - Money & Online Transactions
> - Technology & Content
> - Physical Health & Safety
> - Sensitive & Risk-flagged

### A.8.2 QUERY GENERATION FOR GENDER

> **Prompt Template:**
> We want to test whether the spoken agent can really understand gender from voice. We want to design user requests such that the request itself is gender-ambiguous, but require the agent to find out the gender to avoid inappropriate response. Avoid using the word: 'or', 'and', 'this'. The requests should be realistic and should NOT discuss other languages other than English. The request should be in oral form suitable for text-to-speech. Give me 100 requests (specific user request, not topic or category names) related to {*topic*} in the following format:

1. user_chat1
2. user_chat2
3. ...

You should give ALL the requests without ending early.

**Topics:**
- Cultural and Religious Advice
- Medical and Health Advice
- Gender-specific Activity
- Fashion, Beauty, and Grooming

### A.8.3  QUERY GENERATION FOR EMOTION

**Prompt Template:**
Can you give me some user chats with you that the chat's content itself is emotion-amgiguous and can be normally voiced by OPPOSITE emotions. Avoid the word: ChatGPT, AI. The request should be realistic, detailed and specific. The request should also be in oral form suitable for text-to-speech. I want the specific user chats instead of the category or topic names. Give me the user request and the two OPPOSITE emotions. Give me 100 requests that can be voiced by both {*emotion1*} and {*emotion2*}, related to {*topic*}, in the following format (do not repeat the demo):

1. The city is planning to build a new mall near my house. (emotion1, emotion2)
2. ...
3. ...

You should give ALL the requests without ending early.

**Topics:**
- Personal Life
- Current Events & News
- Entertainment
- Hobbies & Interests
- Work & Studies
- Internet Culture
- Travel
- Food & Drinks
- Relationships
- Technology & Gadgets
- Health & Fitness
- Education
- Finance & Money
- Fashion & Style
- Life Advice
- Cultural Differences
- Dreams & Sleep
- Holidays & Celebrations
- Childhood Memories
- Future Plans

### A.8.4    QUERY GENERATION FOR SARCASM

**Prompt Template:**
You are generating user chat requests that, in plain text, read as entirely positive, cheerful, or supportive, yet could plausibly be sarcastic depending on tone of voice or context. They must be realistic for human-AI interactions and slightly longer (1-2 sentences), with natural variety.

Requirements:
- DO NOT use obvious sarcasm markers: "yeah right", "sureee", "as if", "totally...", ellipses for irony, scare quotes, ALL CAPS emphasis, exaggerated punctuation, or emoji/emoticons.
- Avoid overly negative words (e.g., "hate", "awful", "broken", "slow", "crash").
- Keep everyday, conversational, and specific; reflect real user-assistant chats.
- All items must relate to the topic: {*topic*}.
- Output EXACTLY 100 items as a numbered list starting at 1. One line per item.

Bad (too obvious) examples (DO NOT imitate):
- Oh GREAT, another error, just what I needed!!!
- Yeah right, your 'amazing' update totally helped.

Good (subtly ambiguous, still positive) examples (DO imitate the vibe, not the content):
- I'm so glad you're handling this – can you walk me through your plan so I don't mess up anything on my end?
- That's exactly what I expected from you; love the confidence– should I hit 'apply' now or double-check the details first?
- Wow, you're really on top of things today! Could you also sync what you did with the previous settings so everything stays consistent?

Now produce the list in this format:
1. <item 1>
2. <item 2>
3. ...

You should give ALL the requests without ending early.

**Topics:**
- Personal Life
- Current Events & News
- Entertainment
- Hobbies & Interests
- Work & Studies
- Internet Culture
- Travel
- Food & Drinks
- Relationships
- Technology & Gadgets
- Health & Fitness
- Education
- Finance & Money
- Fashion & Style
- Life Advice
- Cultural Differences

> - Dreams & Sleep
> - Holidays & Celebrations
> - Childhood Memories
> - Future Plans

### A.8.5 SCORING GUIDELINE

This is the scoring guideline designed by the human annotation expert, $r$, in Section 3.2. The user information includes the transcription as well as the emotion, sarcasm, gender, and age labels. The agent information includes the response transcription and a description of the speaking tone.

> You are a human dialogue expert. You will be given a pair of user request audio and an agent response audio. Please focus on the user's speech attributes, including content, emotion, age, gender, sarcasm, and decide whether the agent's response is a good fit in a natural dialogue. Here is a guideline defining the good response of an agent. You need to score the response's quality for Continuation Fitness.
>
> General guideline of Continuation Fitness:
> - MOS on how well the response speech fits the user speech.
> - Listen carefully to spoken content and speaking style of the user speech and the response speech. Decide whether the response speech, including how and what the agent says, is a natural continuation to the user's spoken content and speaking style (emotion, age, gender, sarcasm). Here are some general rules and examples:
>     1. The agent should be friendly, helpful, and considerate, with high EQ.
>     2. Besides the replying content, please also listen carefully to the response's speaking style: emotion, tone, volume and speed, and decide whether the speaking style is appropriate.
>     3. The agent should be aware of the user's emotions to provide emotional companionship. For example, when the user is happy, the agent can share that happiness; when the user is sad, the agent should be sad and empathic.
>     4. If the user is a child, the agent should respond with caution and ensure safety. For example, it should redirect the user if he or she attempts to purchase alcohol online.
>     5. The agent should be aware of the user's gender and personalize responses. For example, it may offer different suggestions for swimwear.
>     6. Suggesting items more related to male for female, or vice versa, is considered very ackward. For example, suggesting masculinised perfume for a female is ackward.
>     7. The agent should recognize sarcasm and understand the user's true intent. For example, when a user gives a sarcastic compliment, the agent should recognize that the underlying sentiment is negative. Hence, it would be weird if the agent feels happy for the compliment.
>
> You should rate the response's quality in 5 points.
> - **5 Points - Perfect (Enhanced):**
>     - The user's age, gender, or speaking style characteristics are recognized and reflected in the response with appropriate enhancements. The user's query contains clear emotional cues, and the response provides empathetic feedback. The user's query has a clear sarcastic tone, and the response offers a high-EQ reassurance or clarification. The user's query is a sincere compliment, and the response is thankful.
>     - **Examples:** When the user is happy, the response shares the joy; when the user is sad, the response offers appropriate comfort. If a minor attempts to purchase alcoholic beverages online, the model provides correct guidance. For a young user, the response uses trendy slang popular among young people. Provides gender-suitable response (i.e. different swimwear suggestions) based on the user's gender. When receiving a sarcastic comment, the model identifies the underlying negative sentiment and responds accordingly.
> - **4 Points - Excellent (No Enhancement):**
>     - The user's paralinguistic cues are addressed so the replying content is good, but the response's vocal tone does not enhance the user's experience.
>     - **Examples:** A neutral-tone response to a female user inquiring about cancer screening. A neutral-tone response to a neutral question. The response's content picks up the user's sarcastic comment, but the tone is not appropriate.

- **3 Points - Average:**
  - The user's paralinguistic cues or other speaking style features are considered, but the response does not provide correct personalized content, though it is not jarring: Mechanical empathy, awkward praise, etc.
  - **Examples:** A happy or sad response to a neutral question.
- **2 Points - Poor:**
  - The user's paralinguistic cues or other speaking style features are considered but poorly addressed. Emotion mismatch: if the agent identifies the wrong user emotion. e.g. Reply to a fearful user as if he/she is sad; Reply to a angry user as if he/she is fearful. Style partially mismatched.
  - **Examples:** A flat response to a sad question. Using slang when responding to an elderly user.
- **1 Point - Very Poor:**
  - The user's paralinguistic cues or other speaking style features are considered but addressed incorrectly. Reverse empathy, condescending tone. e.g. Reply to a sad user as if he/she is happy; Reply to a happy user as if he/she is sad. Completely mismatched style, e.g., responding to an adult in a completely childish tone. Misinterpret a sincere compliment as a negatvie comment, and give apologetic, clarifying, or reassuring comment. Misinterpret a sarcastic compliement as sincere, and give positive or thankful comment.
  - **Examples:** A cheerful response to a sad user. Using language that is too complex for a child. Giving male-specific recommendation to a female, or vise versa.

The information of the user:

{*transcription*}{*emotion*}{*sarcasm*}{*age*}{*gender*}

Here is the information of the agent:

{*transcription*}{*tone*}

Please give the 5-point score and a VERY brief reason in the format: The reason is _; The score is _ .

