# OpenReview forum: "ParaS2S: Benchmarking and Aligning Spoken Language Models for Paralinguistic-aware Speech-to-Speech Interaction"
_ICLR.cc/2026/Conference — ICLR 2026 Poster_

### Official Review · Reviewer_cPKU · 2025-10-26

**Soundness:** 3
**Presentation:** 3
**Contribution:** 3
**Rating:** 6
**Confidence:** 4

**Summary:**

This paper introduces a novel reinforcement learning (RL) framework to enhance the capability of paralinguistic cues for speech-to-speech (S2S) dialogue model. It clearly introduce the topic of natural speech communication which conveying more than just words but also paralinguistic cues such as emotions, tone, speaker attributes and how it impacts the speech response. To address this, the author constructs a benchmark which can automatically evaluates the S2S model as ParaS2SBench for content and style appropriateness, then with the benchmark system, it can use RL approach like GRPO to improve the response content and style fitness.

The major contributions lies on the concrete description of this topic, construct appropriate S2S evaluation as benchmark, demonstration of RL and SFT effectiveness and cost via experiments. The benchmark is characterized by imaginative design and rigorous focus on its objectives, while the exploration of the experiments are also inspiring.

**Strengths:**

1). The construction of this benchmark system is quite clear and scientific. It designs various of domain for different query, contrasting speaking style, and the most key part is the scenario-controlled queries which is designed to control the neutral text content filtering those doesn't convey too much additional paralinguistic information. According to the appendix table for query example and prompt, the data curation is full of authors' thought on this topic and looks very interesting.

2）In the experiments part, besides the validation of the effective of RL framework and SFT analysis, it propose so many questions with experiments which are very instructive. Such as the amount of annotations can RL save? should we invest more costs on SFT or reword model, how is the generalization to real speech? These are realistic and serious problems for this S2S model need to handle and could be continuous and broad interest for the research problem.

**Weaknesses:**

The most weakness of this paper from my perspective is the depth of the experiments part. The author provides rich and many good inspects questions here which is good, however, it seems there isn't any experiment or analysis which shows the most important and critical point view from the author. It may weaken the paper's persuasiveness and confuse reader about the core information obtained from the experiments.

**Questions:**

It would be much friendly if the author could select the most crucial experiments to introduce more profoundly, analysis/conclude them clearly in the main paper session 5 and put the other experiments details in the appendix paper. Anyway, there are some part which is not fully introduced due to the length to the article and need to read the appendix to get the full results.
For the generalization to real speech experiment, is there any human evaluation to compare with the bench score? This is quite curial to this RL framework.

---

> ### Author Response · Authors · 2025-11-21
> **Author Response**
>
> We thank the reviewer for the positive feedback on our work. We address the weaknesses and the raised questions in the following.
>
> > The most weakness of this paper from my perspective is the depth of the experiments part. The author provides rich and many good inspects questions here which is good, however, it seems there isn't any experiment or analysis which shows the most important and critical point view from the author. It may weaken the paper's persuasiveness and confuse reader about the core information obtained from the experiments. It would be much friendly if the author could select the most crucial experiments to introduce more profoundly, analysis/conclude them clearly in the main paper session 5 and put the other experiments details in the appendix paper.
>
> We thank the reviewer for the insightful comment! In fact, the questions in Section 5 are all closely related to thoroughtly and solidly convey only **two core messages**: **(1)** *With proper design, automatic benchmark scoring can align with human judgement* **(2)** *With the benchmark scoring feedback, RL does increase performance compared to SFT across various scenarios*. These two messages are our core contributions, and we verify these contributions by answering a series of inspect questions in Section 5. However, we acknowledge that without a clear context, it could be vague how these inspect questions are related. To resolve this, in the latest revision, we add a new *Overview* paragraph in the begining of Section 5 to illustrate what are we going to do in each subsection any why. Please have a look in the following. Please let us know if this resolve the issue.
>
> ***Overview.** In this section, we aim to answer two core research questions: (1) Can the automatic benchmark scoring reflect human judgments and serve as an effective evaluation tool? and (2) Can RL truly improve performance compared to SFT for this problem? Section 5.1 provides analyses and answers to the first question and show positive correlation. To thoroughly address the second question, we verify it across several subsections. Section 5.2 demonstrates that RL consistently leads to better performance regardless of the amount of SFT warm-up data, supporting our main claim. Section 5.3 points out that only a few hours of SFT data are sufficient to bootstrap the self-improvement process, indicating that RL indeed helps mitigate the data scarcity of paralinguistic-aware S2S. After validating the effectiveness of the RL algorithm, we move on to the practical concern of data construction cost. Section 5.4 provides best practices for balancing the budget allocation between SFT data and RL data, demonstrating the cost efficiency of the proposed framework. Next, we validate the generalizability of the training framework to real speech in Section 5.5, which is essential as the previous experiments are conducted on synthetic speech. Finally, Section 5.6 verifies that our training framework leads to a SOTA model in paralinguistic-aware dialogue by comparison with existing models. These subsections jointly gauge the effectiveness, generalizability, and cost efficiency of the proposed RL framework.*
>
>
> > For the generalization to real speech experiment, is there any human evaluation to compare with the bench score? This is quite curial to this RL framework.
>
> Thank you for the comment! We do conduct human evaluation on the real speech experiments. Please see the right columns in Appendix *A.9 Human Evaluation*. We show the table in the following:
>
> | Model               | IEMOCAP | MELD  | Avg   |
> |---------------------|---------|-------|-------|
> | **Baseline**        |         |       |       |
> | Whisper-GPT-TTS     | 3.601   | 3.552 | 3.487 |
> | **Closed Source**   |         |       |       |
> | GPT-4o voice mode   | 3.862   | 3.694 | 3.778 |
> | **Open Source**     |         |       |       |
> | Qwen2.5 Omni        | 3.713   | 3.581 | 3.647 |
> | GLM 4               | 3.521   | 3.325 | 3.423 |
> | Kimi-Audio          | 2.231   | 2.272 | 2.252 |
> | **Ours**            |         |       |       |
> | Kimi-Audio SFT      | 4.212   | 3.407 | 3.810 |
> | Kimi-Audio GRPO     | 4.336   | 3.859 | 4.098 |
> | **Topline**         |         |       |       |
> | GPT-TTS             | 4.911   | 4.925 | 4.918 |
>
> The results demonstrate that our RL framework significantly outperforms the original Kimi-Audio and the existing models under the subjective evaluation.

---

> > ### Author Response · Authors · 2025-11-21
> > **Thank you for your thoughtful review**
> >
> > Finally, we understand that it takes considerable time to thoroughly read and review a paper. We sincerely thank the reviewer for spending the time on both the review and the rebuttal, and we hope you find our responses helpful. If you find our reply helpful, we kindly invite you to raise the score, although we will fully respect your decision if you choose to maintain the original judgment. Thanks!
> >
> > Sincerely, ParaS2S authors

---

### Official Review · Reviewer_qCT1 · 2025-10-29

**Soundness:** 2
**Presentation:** 2
**Contribution:** 2
**Rating:** 4
**Confidence:** 4

**Summary:**

This paper introduces ParaS2S, a reinforcement learning framework that enables speech-to-speech models to generate responses with appropriate content and speaking style. Using the new ParaS2SBench for automatic content and style evaluation, ParaS2S improves paralinguistic awareness by 11% over supervised fine-tuning while requiring far fewer annotations.

**Strengths:**

The motivation to evaluate paralinguistic responses in speech language models is both natural and important. This reviewer appreciates the authors’ effort in advancing research on this topic. Moreover, the overall workload presented in this paper appears substantial.

**Weaknesses:**

(1) The core contributions of this paper are somewhat unclear. It mainly includes two parts: a new benchmark for evaluating speech response style and content, and an alignment technique for tuning speech language models. However, each contribution appears incomplete on its own. The benchmarking part omits many relevant speech and speech-to-speech models, while the proposed alignment method lacks sufficient novelty and empirical validation.

(2) The citation format should follow the ICLR template by using \citep instead of \cite, as the current style blends citations into the text and reduces readability.

(3) Table 1 reports several numerical results to show that the evaluation aligns with human judgments, but the justification is not rigorous. The paper should clarify what criteria define “closeness” to human evaluation and why they are reasonable. For instance, in the Emotion S2S model, the gap between GPT and human evaluations does not appear negligible.

(4) The novelty of the alignment approach is limited. If the paper argues for the necessity of GRPO, this claim should be empirically supported, and comparisons with existing methods such as SpeechAlign are essential.

(5) The GRPO alignment is evaluated only on the Kimi-Audio base model. A more comprehensive study should include multiple base models to demonstrate that the proposed strategy generalizes beyond a specific setup.

In conclusion, while the paper presents a decent amount of work, it remains incomplete by publication standards. The authors are encouraged to focus on a single, well-developed contribution—either the benchmarking framework or the alignment technique.

**Questions:**

(1) What is the primary contribution of this paper — the new benchmark or the proposed alignment strategy? The authors should clarify which aspect represents the core focus of the work.

(2) Why is the proposed alignment strategy not compared with SpeechAlign? The two methods appear quite similar, except for the adoption of the GRPO technique. Since SpeechAlign also employs DPO and evaluates comparable aspects of speech language models, a direct comparison is necessary to highlight the distinction and advantage of the proposed approach.

---

> ### Author Response · Authors · 2025-11-21
> **Author Response (1/4)**
>
> We sincerely thank the reviewer for the helpful comment. We gratefully accept some comments and wish to further discuss and clarify a few points. We realize that some materials are essential and should be placed in main text instead of appendix to avoid misunderstanding (e.g. correlation/closeness analysis between benchmark scores and human judgement) and we have provided the revised version. We invite the reviewer to discuss several matters in the following. Thanks!
>
> ### Regarding the core contribution
>
> > The core contributions of this paper are somewhat unclear. It mainly includes two parts: a new benchmark for evaluating speech response style and content, and an alignment technique for tuning speech language models.
>
> > What is the primary contribution of this paper — the new benchmark or the proposed alignment strategy? The authors should clarify which aspect represents the core focus of the work.
>
> Thanks for the comment! Regarding the contribution, we would like to emphasize that our contribution does not lie solely in technical novelty (i.e. RL method), but in *defining and formulating a new problem* that has not been previously discussed: **paralinguistic-aware speech-to-speech interaction**. Because this is a new problem, the field lacks essential components such as datasets, evaluation protocols, and baseline models. Our work takes the first step by *contributing all of these components together*. From this perspective, our contribution can be viewed as a **benchmark platform for the new problem** that includes the dataset, evaluation tools, and several strong baseline models (SFT and RL) surpassing all existing models.
>
> We thank the reviewer for pointing out the ambiguity of our position. We have revised the *Abstract* and *Introduction* to claim the SFT and GRPO methods as strong baselines in the benchmark ecosystem instead of our major algorithmic proposal.
>
> ### Regarding the insufficient baselines
>
> > The benchmarking part omits many relevant speech and speech-to-speech models
>
> > Why is the proposed alignment strategy not compared with SpeechAlign? The two methods appear quite similar, except for the adoption of the GRPO technique. Since SpeechAlign also employs DPO and evaluates comparable aspects of speech language models, a direct comparison is necessary to highlight the distinction and advantage of the proposed approach.
>
> We report baseline results for several representative SOTA speech-to-speech models such as GPT-4o Voice Mode, Gemini, Kimi-Audio, and Qwen2.5-Omni, and we demonstrate their weaknesses. We would be happy to include additional relevant models—if the reviewer could point out which important models we may have missed, we would gladly add new benchmark results. We also note that the mentioned SpeechAlign model is *not* a speech-to-speech model; it is a language-model-based TTS system (similar to VALL-E), improved by DPO. Therefore, it does not have dialogue capabilities and is not an appropriate comparison target. Additionally, although the SpeechAlign authors provide a codebase link, the code and model are not actually open-sourced.

---

> > ### Author Response · Authors · 2025-11-21
> > **Author Response (2/4)**
> >
> > ### Regarding the novelty, necessity and emperical validation of our RL approach
> >
> > > The proposed alignment method lacks sufficient novelty and empirical validation.
> >
> > > The novelty of the alignment approach is limited. If the paper argues for the necessity of GRPO, this claim should be empirically supported, and comparisons with existing methods such as SpeechAlign are essential.
> >
> > We acknowledge the lack of novelty in the RL algorithm. As outlined in the previous answer, we do not aim for a novel method (technical contribution is not our main contribution); instead, we study the most standard, simple, yet effective approach to achieve SOTA paralinguistic-aware dialogue capabilities, serving as strong baselines in our benchmark platform. As a result, we try the most straightforward SFT, and we find RL even more suitable for this specific problem.
> >
> > In fact, we do not argue for the necessity of GRPO; we choose to proceed with GRPO simply due to its implementation simplicity, as we find constructing a large-scale offline preference dataset for DPO too costly and time-prohibitive. Therefore, we choose to learn a distilled, lightweight reward model to conduct the preference labeling process on-the-fly with a smaller model size. This is a consideration regarding implementation simplicity rather than a comparison of RL method effectiveness. As a result, we show that RL mitigates the data-scarcity issue of this problem by realizing it with GRPO, and we hope future studies can move forward by replacing GRPO with other, better RL algorithms.
> >
> > Comparing to existing RL methods is required to validate our approach *if* there is an existing RL baseline for this problem. However, we would like to highlight that there is *no* existing RL method for this problem (paralinguistic-aware S2S). *It was unknown whether RL works for this problem and how to design the reward.* The mentioned SpeechAlign is an RL method for *language-model-based TTS systems (like VALL-E)*, which do not have dialogue capabilities and are very different from what we are studying, including the design of the preference signal. Instead of determining the valid baseline RL algorithm, we would like to clarify that our contribution does not lie in *proposing GRPO or any specific RL algorithm*, but in *being the first to advocate the use of RL for paralinguistic-aware S2S*. GRPO is simply one realization method. Theoretically, any RL algorithm used would constitute our contribution as long as it works, and it would be the first RL method/baseline verified to work for this problem.
> >
> > In conclusion, we respectfully thank the reviewer for the comment, although we believe there might be some miscommunication that can be easily resolved with some discussion. Since SpeechAlign works on a completely different problem and is not a valid baseline, it is in fact inappropriate to compare with it. Also, since our main argument is *verifying RL is helpful with proper reward design*, the baseline should be SFT instead of other RL algorithms.
> >
> > **P.S.** Regarding the experimental validation for the effectiveness of GRPO, we also provide several ablation studies in the Appendix, such as *(A.5) GRPO batch size, group size, and KL regularization weight*, *(A.8) Generalizability to real speech after GRPO*, and *(A.10) Whether GRPO hurts the original capabilities*. We hope these experiments help gauge the effectiveness of GRPO on this problem. Please let us know if any critical ablation is missing.

---

> > > ### Author Response · Authors · 2025-11-21
> > > **Author Response (3/4)**
> > >
> > > ### Regarding the closeness between the benchmark scores and human judgement
> > >
> > > > Table 1 reports several numerical results to show that the evaluation aligns with human judgments, but the justification is not rigorous. The paper should clarify what criteria define “closeness” to human evaluation and why they are reasonable. For instance, in the Emotion S2S model, the gap between GPT and human evaluations does not appear negligible.
> > >
> > > We thank the reviewers for the comments. In fact, we do define “closeness” to human evaluation as the **Pearson correlation between automatic scores and human scores**, which is reasonable, as Pearson correlation is also widely used in NLP when studying the alignment between LLM judgments and human judgments [1] to gauge the effectiveness of LLM-as-a-judge. We understand the reviewer’s misunderstanding may stem from the fact that we only briefly mentioned this study in one sentence due to the page limit:
> > >
> > > *“Finally, correlation analysis (Appendix A.7) shows scores between human experts and the benchmark pipeline are strongly correlated (above 0.7) across paralinguistic categories, all statistically significant.”*
> > >
> > > In Appendix A.7, we report the correlation table between human experts and the benchmark pipeline by collecting numerous query–response pairs and labeling each pair with both a human fitness score and a benchmark-predicted score:
> > >
> > > |              | Age   | Emotion | Gender | Sarcasm | All   |
> > > |--------------|-------|---------|--------|---------|-------|
> > > | **Pearson r**| 0.862 | 0.760   | 0.702  | 0.779   | 0.773 |
> > > | **Kendall τ**| 0.532 | 0.475   | 0.421  | 0.451   | 0.468 |
> > >
> > > All the *p*-values are smaller than 0.05, showing statistical significance. The results for both Pearson and Kendall correlation support that our benchmark pipeline is *close* to human judgment *to a high degree*. We further include Kendall correlation in this rebuttal for completeness due to its use in [1] and [2]. Pearson correlation measures linear correlation, while Kendall correlation measures rank correlation. Note that Kendall coefficients are usually much lower than Pearson coefficients, and values above 0.4 are often considered strong [1,2].
> > >
> > > In the latest revision, we have moved the correlation study to the main text for completeness.
> > >
> > > We acknowledge that in Table 1, the Emotion S2S results show that *humans gave lower absolute scores compared to benchmark scores*. However, a closer examination reveals that the **ranking across models is preserved**, as is the ranking among query–response pairs, supported by the high Kendall τ score. This indicates that our benchmark scoring may not always match the exact absolute scores assigned by human experts, but their relative ordering is highly similar. Therefore, we believe the benchmark pipeline still provides reasonable and useful judgments—at least sufficiently reliable to serve as feedback for improving models via RL.
> > >
> > > Finally, we emphasize that we do not rely solely on the benchmark scoring as the final judgment when comparing models. We additionally conduct human evaluations to ensure that RL with this scoring feedback indeed leads to improvement. These results are provided in Appendix A.9.
> > >
> > >
> > > [1] Cheng-Han Chiang and Hung-yi Lee. 2023. A Closer Look into Using Large Language Models for Automatic Evaluation. In Findings of the Association for Computational Linguistics: EMNLP 2023, pages 8928–8942, Singapore. Association for Computational Linguistics.
> > > [2] Cheng-Han Chiang and Hung-yi Lee. 2023. Can Large Language Models Be an Alternative to Human Evaluations?. In Proceedings of the 61st Annual Meeting of the Association for Computational Linguistics (Volume 1: Long Papers), pages 15607–15631, Toronto, Canada. Association for Computational Linguistics.

---

> ### Author Response · Authors · 2025-11-21
> **Author Response (4/4)**
>
> ### Regarding the generalizability of the training pipeline
>
> > The GRPO alignment is evaluated only on the Kimi-Audio base model. A more comprehensive study should include multiple base models to demonstrate that the proposed strategy generalizes beyond a specific setup.
>
> We thank the reviewer for the valuable suggestion. We chose Kimi-Audio because it is the SOTA open-source S2S model, and it adopts the standard decoder-only architecture like other S2S models, making studies on this specific model representative. We also attempted to apply the training pipeline to another SOTA S2S model like Qwen-2.5 Omni, but we found that they do not release the speech tokenizer, making further fine-tuning of the released model infeasible. We acknowledge that applying the training pipeline to other S2S models would strengthen the generalizability of our method. However, we also admit that during the rebuttal period we do not have the resources to conduct such additional training, as the internship period has already ended and many open-source solutions are incomplete (only releasing the model without the necessary speech tokenizer). We recognize that this weakness exists and is difficult to address at the moment. We hope to acquire the necessary resources in the future to extend the study to other models and release the results alongside the current materials. Thanks!
>
> ### Regarding the presentation format
>
> > The citation format should follow the ICLR template by using \citep instead of \cite, as the current style blends citations into the text and reduces readability.
>
> Thank you very much for pointing out! We did not notice this formatting error. It is fixed in the new revision. Thank you!
>
> ### Conclusing Remarks
>
> Finally, we fully understand the reviewer’s points and thank the reviewer for the valuable feedback. We hope the reviewer can re-evaluate our contribution after the clarification and raise the score, but we will still respectfully accept the results if the reviewer maintains the original judgment. Thanks!

---

> > ### Author Response · Authors · 2025-11-21
> > **Thank you for your thoughtful review**
> >
> > Finally, we understand that it takes considerable time to thoroughly read and review a paper. We sincerely thank the reviewer for spending the time on both the review and the rebuttal, and we hope you find our responses helpful. If you find our reply helpful, we kindly invite you to raise the score, although we will fully respect your decision if you choose to maintain the original judgment. Thanks!
> >
> > Sincerely, ParaS2S authors

---

> > > ### Comment · Reviewer_qCT1 · 2025-11-27
> > > **Replying to Authors**
> > >
> > > Thank you for your rebuttal. I would like to share my feedback below:
> > >
> > > The primary concern lies in the clarity of writing and the depth of the contribution. The current presentation lacks sufficient structure, which makes it difficult for readers to clearly identify the core message of the paper. This sentiment appears to be echoed by other reviewers. Additionally, as mentioned in the rebuttal, the paper introduces a novel task and feature for benchmarking and evaluation. Given this novelty, a clear and informative introduction to paralinguistic features is essential. At present, it remains unclear what these features specifically entail and how the proposed benchmark is designed to assess a model’s capability in this regard. This topic should be systematically and thoroughly elaborated in the introduction section.
> > >
> > > Furthermore, I strongly recommend that the authors initially focus on either alignment or benchmarking, as each represents a substantial research direction with considerable depth. From the rebuttal, I understand that alignment algorithms are not the primary focus of the paper; however, the manuscript presents alignment at the same conceptual level as benchmarking, which risks shifting attention away from the main contribution. While I appreciate that running foundation models is resource-intensive and exploratory work can be valuable, the current version does not demonstrate sufficient depth in either direction.
> > >
> > > Overall, I regret to say that I will maintain my score at this moment.

---

### Official Review · Reviewer_hX74 · 2025-10-31

**Soundness:** 3
**Presentation:** 3
**Contribution:** 3
**Rating:** 6
**Confidence:** 4

**Summary:**

The study introduces a methodology for enabling Speech-to-Speech (S2S) language models to recognize and respond to critical paralinguistic elements, such as emotion, intonation, and speaker characteristics, which extends far beyond simple content transmission.

The authors present two core components:

* **ParaS2SBench:** An automated benchmark designed to evaluate how effectively S2S models align with both the **content** and **style** of an utterance.
* **ParaS2SAlign:** A learning framework that utilizes Reinforcement Learning (RL) to achieve model alignment directly at the waveform level.

The benchmark scores show a high correlation (>0.7) with human evaluations and the RL approach achieves a notable 11% performance improvement over Supervised Fine-Tuning (SFT), alongside a five-fold enhancement in label efficiency.

**Strengths:**

- The paper puts forward a novel benchmark and dataset, with a welcome commitment to their public release.
- The authors provide a valuable analysis of the respective impacts of RL and SFT on the modeling of non-verbal conversational aspects within the proposed framework.

**Weaknesses:**

* **On the Reward Model:** A point of consideration emerged regarding the reward model. I'm respectfully curious about the potential for it to be somewhat overfitted to the specific synthesis engines used for evaluation, namely the GPT-based TTS and CosyVoice. I would be interested to hear the authors' perspective on its generalization capabilities to other speech styles.
* **On Data Synthesis:** Additionally, as the audio corresponding to the evaluated scenarios appears to be entirely synthetic, a slight query arises regarding potential constraints on the diversity and complexity of the model's expressive output. I wonder if this might impact the model's ability to capture the full spectrum of nuances present in organic, human-to-human interaction.

**Questions:**

1.  **Readability:** I noticed a minor formatting point where the inconsistent use of parentheses for citations occasionally impacted readability. Clarifying this convention throughout the manuscript might be beneficial for readers.
2.  **Data Composition:** Could the authors please specify the total number of distinct speakers represented in the training data and the ParaS2SBench benchmark, respectively?
3.  **Confidence Intervals:** I would find it very helpful to see the 95% confidence intervals for the reported GPT and human evaluation scores, as this would further strengthen the statistical significance of the findings.
4.  **Performance on Existing Capabilities:** Finally, a point of great interest is the trade-off with existing abilities. I would be grateful if the authors could provide an analysis of any performance degradation on foundational capabilities (e.g., as measured by VoiceBench) after the application of SFT and, subsequently, the full RL alignment process.

---

> ### Author Response · Authors · 2025-11-21
> **Author Response (1/3)**
>
> We thank the reviewer for the positive feedback on our work. We address the weaknesses and the raised questions in the following.
>
> > **On the Reward Model:** A point of consideration emerged regarding the reward model. I'm respectfully curious about the potential for it to be somewhat overfitted to the specific synthesis engines used for evaluation, namely the GPT-based TTS and CosyVoice. I would be interested to hear the authors' perspective on its generalization capabilities to other speech styles.
>
> We thank the reviewer for the insightful comment. We also share this concern. To study generalizability, we conduct tests on real speech, outlined in Appendix *A.8 Generalizability to Real Speech*. We use real speech from the emotion datasets IEMOCAP and MELD.
>
> | Method                       | IEMOCAP test | MELD test | Average |
> |-----------------------------|--------------|-----------|---------|
> | GRPO on IEMOCAP+MELD        | 4.394        | 3.947     | 4.166   |
> | GRPO on MELD                | 4.386        | 3.942     | 4.164   |
> | GRPO on IEMOCAP             | 4.356        | 3.872     | 4.114   |
> | SFT+GRPO on Synthetic Data  | 4.258        | 3.349     | 3.803   |
> | SFT on Synthetic Data       | 4.121        | 3.307     | 3.714   |
> | Base Model (Kimi-Audio)     | 1.365        | 1.166     | 1.265   |
>
> First of all, we can observe that SFT on synthetic data leads to huge performance improvements on both IEMOCAP and MELD. RL on synthetic data also brings improvements on both datasets. On the other hand, we find that further applying GRPO on real datasets still slightly improves performance. These results highlight that learning from synthetic data generalizes to real speech (in a zero-shot fashion) to a large extent, while further in-domain training still provides additional gains.
>
> We verified the generalizability to real speech on emotion, but it remains unexplored for the other three paralinguistic features. However, we find that existing datasets are mostly *paralinguistic-irrelevant*. That is, the best responses to the spoken content of the utterances are usually irrelevant to the paralinguistic feature. For example, asking about the current U.S. president with either a male or female voice should yield the same answer. As a result, we only study generalizability to real speech for emotion at the current stage.
>
> In conclusion, we test the generalization capabilities and find that the model performs well at least in the emotion category, and we hope future conversational datasets for other speech types will allow more comprehensive evaluation. Thanks for the critical and helpful comment. We have added this discussion in to the new *Limitation* section.
>
> > **On Data Synthesis:** Additionally, as the audio corresponding to the evaluated scenarios appears to be entirely synthetic, a slight query arises regarding potential constraints on the diversity and complexity of the model's expressive output. I wonder if this might impact the model's ability to capture the full spectrum of nuances present in organic, human-to-human interaction.
>
> We thank the reviewer for the valuable feedback. We acknowledge that the synthetic responses indeed affect the diversity and expressiveness of the model’s output. This issue stems from the lack of paralinguistic-aware dialogue datasets. As a result, creating the suitable agent responses ourselves is currently the only solution, and our model’s diversity and expressiveness are thus upper-bounded by the TTS system, namely OpenAI *gpt-4o-mini-tts*. Despite the TTS system already showing a high degree of naturalness and expressiveness, we observe that its style controllability is quite unstable, and the generated speech sometimes does not adhere to the style prompt, which affects the training data quality.
>
> Consequently, to approach a more human-like, empathetic, and expressive spoken agent, a natural next step would be to collect real speech dedicated to this problem for both the training and testing sets, including the input queries and response demonstrations, to fully capture the nuances in organic, human-to-human interactions. We highly appreciate the reviewer for pointing out this limitation, and we have added the discussion to the new *Limitation* section.
>
> > **Readability:** I noticed a minor formatting point where the inconsistent use of parentheses for citations occasionally impacted readability. Clarifying this convention throughout the manuscript might be beneficial for readers.
>
> Thanks again for the constructive feedback! Another reviewer also points out this presentation issue. We have fixed this issue in the latest revision.

---

> > ### Author Response · Authors · 2025-11-21
> > **Author Response (2/3)**
> >
> > > **Data Composition:** Could the authors please specify the total number of distinct speakers represented in the training data and the ParaS2SBench benchmark, respectively?
> >
> > We use *CosyVoice2* for synthesizing gender and age, and OpenAI *gpt-4o-mini-tts* for emotion and sarcasm. We try to maximize speaker diversity for gender and age. We sample speech prompts from diverse speakers—male, female, and adult—from CommonVoice and LibriSpeech. For child speech, we leverage NNCES [1]. We report the total speaker counts for the training/testing split in the following table:
> >
> > | Category | Train set | Test set |
> > |----------|-----------|----------|
> > | Gender   | 3357      | 1200     |
> > | Age      | 2302      | 625      |
> > | Emotion  | 6         | 5        |
> > | Sarcasm  | 6         | 5        |
> >
> > As shown in the table, the speakers are highly diverse for gender and age. On the other hand, *gpt-4o-mini-tts* has a limitation in that only 11 speakers are supported. We tested whether *CosyVoice2* can clone speaker timbre along with emotional expression. However, we find that the resulting emotional cues are very vague and cannot pass the human quality test. As with other SOTA ICL-based TTS systems, they are not expressive or high-fidelity enough for emotional or sarcastic speech. As a result, we stick to *gpt-4o-mini-tts* for expression correctness. Still, we enforce a disjoint training/testing speaker split to avoid our trained model overfitting to specific speakers.
> >
> > [1] https://www.kaggle.com/datasets/kodaliradha20phd7093/nonnative-children-english-speech-nncescorpus
> >
> > > **Confidence Intervals:** I would find it very helpful to see the 95% confidence intervals for the reported GPT and human evaluation scores, as this would further strengthen the statistical significance of the findings.
> >
> > Thanks for the helpful comment. We provide the 95% CI for both the GPT scores and human evaluation scores in the following table. As shown by the table, we approaches, including SFT and GRPO, significantly outperform the original Kimi-Audio and the existing models.
> >
> > | Model               | GPT Synthetic [CI]        | GPT Real [CI]            | Human Synthetic [CI]        | Human Real [CI]            |
> > |---------------------|---------------------------|---------------------------|-------------------------------|-----------------------------|
> > | **Baseline**        |                           |                           |                               |                             |
> > | Whisper-GPT-TTS     | [2.872, 3.172]            | [3.307, 3.667]            | [2.962, 3.242]                | [3.317, 3.657]              |
> > | **Closed Source**   |                           |                           |                               |                             |
> > | GPT-4o Voice mode   | [3.164, 3.404]            | [3.509, 3.769]            | [3.347, 3.567]                | [3.648, 3.908]              |
> > | Gemini              | [3.317, 3.577]            | [3.612, 3.912]            | —                             | —                           |
> > | **Open Source**     |                           |                           |                               |                             |
> > | Qwen2.5 Omni        | [3.118, 3.378]            | [3.452, 3.772]            | [3.363, 3.603]                | [3.507, 3.787]              |
> > | GLM 4               | [2.873, 3.193]            | [2.887, 3.187]            | [2.994, 3.274]                | [3.263, 3.583]              |
> > | LLaMA-Omni 2        | [3.095, 3.335]            | [3.303, 3.583]            | —                             | —                           |
> > | Freeze-Omni         | [2.510, 2.850]            | [2.748, 3.148]            | —                             | —                           |
> > | Kimi-Audio          | [2.742, 3.042]            | [1.025, 1.505]            | [2.764, 3.084]                | [2.062, 2.442]              |
> > | **Ours**            |                           |                           |                               |                             |
> > | Kimi-Audio SFT      | [3.996, 4.156]            | [3.634, 3.794]            | [4.009, 4.169]                | [3.710, 3.910]              |
> > | Kimi-Audio GRPO     | [4.381, 4.501]            | [4.091, 4.231]            | [4.337, 4.477]                | [4.018, 4.178]              |
> > | **Topline**         |                           |                           |                               |                             |
> > | GPT-TTS             | [4.655, 4.755]            | [4.706, 4.826]            | [4.794, 4.894]                | [4.858, 4.978]              |

---

> > > ### Author Response · Authors · 2025-11-21
> > > **Author Response (3/3)**
> > >
> > > > **Performance on Existing Capabilities:** Finally, a point of great interest is the trade-off with existing abilities. I would be grateful if the authors could provide an analysis of any performance degradation on foundational capabilities (e.g., as measured by VoiceBench) after the application of SFT and, subsequently, the full RL alignment process.
> > >
> > > Thanks for the interest! We already included such studies in the first submission. Please check Appendix *A.6 Ablation for GRPO Training* and *A.10 Intelligence Analysis*. In short, GRPO requires proper tuning of the KL regularization weight to learn paralinguistic-aware interaction while preserving the original capabilities. It is indeed feasible to acquire SOTA paralinguistic-aware dialogue capabilities with minimal drop in the original capabilities. We quantify the original capabilities using VoiceBench, which includes daily QA, knowledge-intensive QA, and instruction-following tasks in both closed-ended and open-ended scenarios. In our final model, the original capabilities basically remain the same, with great improvement in paralinguistic-aware interaction. This is achieved by a lightweight SFT (only 2 epochs on the SFT data) and a KL-regularized GRPO. Empirically, we find that optimizing SFT for too many epochs leads to notable degradation of the original capabilities compared to our "warmup -> GRPO" approach.

---

> > > > ### Author Response · Authors · 2025-11-21
> > > > **Thank you for your thoughtful review**
> > > >
> > > > Finally, we understand that it takes considerable time to thoroughly read and review a paper. We sincerely thank the reviewer for spending the time on both the review and the rebuttal, and we hope you find our responses helpful. If you find our reply helpful, we kindly invite you to raise the score, although we will fully respect your decision if you choose to maintain the original judgment. Thanks!
> > > >
> > > > Sincerely, ParaS2S authors

---

> > > > > ### Comment · Reviewer_hX74 · 2025-11-26
> > > > >
> > > > > Thank you for your response. I apologize for overlooking the VoiceBench results in the appendix. I will keep my rating unchanged.

---

### Official Review · Reviewer_zJW8 · 2025-11-01

**Soundness:** 3
**Presentation:** 3
**Contribution:** 3
**Rating:** 6
**Confidence:** 3

**Summary:**

This paper introduces an innovative benchmark and reinforcement learning framework for paralinguistic-aware speech-to-speech (S2S) models, effectively addressing the current limitations in handling paralinguistic information such as emotion, tone, gender, and age. The authors design an automated data curation and speech synthesis pipeline, and leverage a reward model for efficient training and evaluation. Experimental results demonstrate that the RL approach achieves significantly better content and style appropriateness than conventional supervised fine-tuning, with much lower data and annotation costs. Overall, this work is forward-looking and practical, providing valuable tools and references for the development of paralinguistic-aware S2S models.

**Strengths:**

1. The proposed paralinguistic-aware S2S reinforcement learning framework is highly practical, effectively enhancing the model's ability to understand and generate paralinguistic information such as emotion and tone, which provides valuable tools and methods for the advancement of speech dialogue systems.
2. The experiments are thoroughly designed, covering various paralinguistic factors and realistic scenarios. The results comprehensively validate the significant improvements in content and style appropriateness achieved by the proposed method, making the findings highly convincing.
3. The paper is well-structured and clearly articulated, with a rigorous logical flow. It progresses coherently from problem background, method design, to experimental validation and result analysis, making it easy for readers to understand and follow.

**Weaknesses:**

1. The methodological innovation of the paper is limited, as it merely applies GRPO in a straightforward manner.
2. The presentation lacks intuitiveness; it is difficult to fully convey the paralinguistic features of audio through text alone. It would be better if there were a demo page or web-based showcase.
3. Some references are missing, such as [1]:
[1] Omnichat: Enhancing spoken dialogue systems with scalable synthetic data for diverse scenarios. arXiv preprint arXiv:2501.01384.

**Questions:**

1. In your data curation and speech synthesis process, how do you ensure that the generated paralinguistic styles (such as emotion, age, gender, etc.) are sufficiently authentic and diverse?
2. After SFT and GRPO, does the model’s original capability decrease compared to the results reported for Kimi-Audio?

---

> ### Author Response · Authors · 2025-11-21
> **Author Response (1/2)**
>
> We thank the reviewer for the positive feedback on our work. We address the weaknesses and the raised questions in the following.
>
> > The methodological innovation of the paper is limited, as it merely applies GRPO in a straightforward manner.
>
> We fully understand your point and acknowledge the limited methodological innovation. In the meanwhile, we would like to highlight that our contribution mainly lies in *defining and formulating a new problem with a new benchmark platform*, consisting of the dataset, evaluation protocols (and tools), and strong baseline models (SFT and RL) that surpass all existing models. From this perspective, despite the limited methodological innovation, our standard, simple, yet effective RL approach paves a practical pathway for future work in acquiring paralinguistic-aware dialogue capabilities, and the constructed benchmark platform could foster future studies on devising new methodologies to improve models.
>
> > The presentation lacks intuitiveness; it is difficult to fully convey the paralinguistic features of audio through text alone. It would be better if there were a demo page or web-based showcase.
>
> Thanks for the feedback! Due to the page limit, we understand the difficulty without concrete examples. We provide text-based examples in *Appendix A.2 ParaS2SBench statistics and examples*. We show the table in the following. The model responses shown in the table are for demonstration and not part of the benchmark dataset.
>
> ---
>
> ## Emotion
> - **User (happy):**
>   The city is planning to build a new mall near my house.
>
>   **Model (cheerful):**
>   That’s wonderful! A new mall will bring so many shops and restaurants to explore—sounds exciting!
>
> - **User (angry):**
>   The city is planning to build a new mall near my house.
>
>   **Model (empathetic):**
>   I'm sorry to hear that you’re upset about it. Would you like to share what concerns you most about the new mall?
>
> ---
>
> ## Sarcasm
> - **User (sincere):**
>   It looks like you’ve outdone yourself with these answers!
>
>   **Model (happy):**
>   Thank you! I'm glad my answers were helpful—let me know if you’d like me to expand on anything.
>
> - **User (sarcastic):**
>   It looks like you’ve outdone yourself with these answers!
>
>   **Model (apologetic):**
>   Hmm, it sounds like you might not be happy with the answers. Could you tell me what was missing or what you’d like done differently?
>
> ---
>
> ## Age
> - **User (adult):**
>   I need instructions on buying stock with my money.
>
>   **Model (confident):**
>   Sure! I can walk you through opening a brokerage account, researching companies, and placing your first order step by step.
>
> - **User (child):**
>   I need instructions on buying stock with my money.
>
>   **Model (supportive):**
>   Investing is exciting! Since you might be under 18, you may need help from a parent or guardian to open a custodial account before you can buy stocks. Would you like me to explain how that works?
>
> ---
>
> ## Gender
> - **User (male):**
>   Help me pick workout clothes for the gym.
>
>   **Model (professional):**
>   Absolutely! Breathable tank tops, moisture-wicking T-shirts, and flexible shorts are great choices. Do you prefer a sporty or casual style?
>
> - **User (female):**
>   Help me pick workout clothes for the gym.
>
>   **Model (professional):**
>   Sure! High-support sports bras, moisture-wicking leggings, and lightweight tops are excellent options. Would you like suggestions for specific brands?
>
> Also, we have prepare the demo page for you to understand the testing queries with contrasting speakeing styles and the model responses. Here is the demo page: https://paras2sbench.github.io/. We hope these examples help clarify our presentation! The demo page link is also provided in the latest revision.

---

> ### Author Response · Authors · 2025-11-21
> **Author Response (2/2)**
>
> > Some references are missing, such as Omnichat: Enhancing spoken dialogue systems with scalable synthetic data for diverse scenarios. arXiv preprint arXiv:2501.01384.
>
> Thank you very much! We have included the dicussion for this related work in the revision.
>
> > In your data curation and speech synthesis process, how do you ensure that the generated paralinguistic styles (such as emotion, age, gender, etc.) are sufficiently authentic and diverse?
>
> We first discuss the diversity, followed by the authenticity.
>
> For diversity, there is spoken content diversity and style diversity. For content diversity, as illustrated in the main text, we generate queries covering diverse topics and scenarios, including interests, work, studies, relationships, travel, health, religion, fashion, finance, and more.
>
> For style diversity, it depends on the TTS engine we use. We use *CosyVoice2* for gender and age, and OpenAI *gpt-4o-mini-tts* for emotion and sarcasm. The former is an in-context-learning (ICL)-based TTS that clones any speaker timbre in a zero-shot fashion; the latter maximizes expressiveness by controlling the speaking style using text prompts. For the style prompts in emotional and sarcastic/sincere speech, we generate a new style prompt for each TTS synthesis, specifying the emotion, tone, volume, speed, and intonation to maximize speaking-style diversity. On the other hand, we maximize speaker diversity for gender and age. We sample speech prompts from diverse speakers—male, female, and adult—from CommonVoice and LibriSpeech. For child speech, we leverage NNCES [1]. We report the total speaker counts for the training/testing split in the following table:
>
> | Category | Train set | Test set |
> |----------|-----------|----------|
> | Gender   | 3357      | 1200     |
> | Age      | 2302      | 625      |
> | Emotion  | 6         | 5        |
> | Sarcasm  | 6         | 5        |
>
> As shown in the table, the speakers are highly diverse for gender and age. On the other hand, *gpt-4o-mini-tts* has a limitation in that only 11 speakers are supported. We tested whether *CosyVoice2* can clone speaker timbre along with emotional expression. However, we find that the resulting emotional cues are very vague and cannot pass the human quality test. As with other SOTA ICL-based TTS systems, they are not expressive or high-fidelity enough for emotional or sarcastic speech. As a result, we stick to *gpt-4o-mini-tts* for expression correctness. Still, we enforce a disjoint training/testing speaker split to avoid our trained model overfitting to specific speakers.
>
> For authenticity, we first conduct a series of model-based filtering, and finally rely on human experts for spoken content and style correctness checks. Three human annotators are recruited, and only the audio whose content and style pass the review of all annotators is kept in the final testing set. Details are outlined in *Section 3.1 Query Mining*.
>
> [1] https://www.kaggle.com/datasets/kodaliradha20phd7093/nonnative-children-english-speech-nncescorpus
>
> > After SFT and GRPO, does the model’s original capability decrease compared to the results reported for Kimi-Audio?
>
> Thanks for the interest! We already included such studies in the first submission. Please check Appendix *A.6 Ablation for GRPO Training* and *A.10 Intelligence Analysis*. In short, GRPO requires proper tuning of the KL regularization weight to learn paralinguistic-aware interaction while preserving the original capabilities. It is indeed feasible to acquire SOTA paralinguistic-aware dialogue capabilities with minimal drop in the original capabilities. We quantify the original capabilities using VoiceBench, which includes daily QA, knowledge-intensive QA, and instruction-following tasks in both closed-ended and open-ended scenarios. In our final model, the original capabilities basically remain the same, with great improvement in paralinguistic-aware interaction.

---

> > ### Author Response · Authors · 2025-11-21
> > **Thank you for your thoughtful review**
> >
> > Finally, we understand that it takes considerable time to thoroughly read and review a paper. We sincerely thank the reviewer for spending the time on both the review and the rebuttal, and we hope you find our responses helpful. If you find our reply helpful, we kindly invite you to raise the score, although we will fully respect your decision if you choose to maintain the original judgment.
> > Thanks!
> >
> > Sincerely,
> > ParaS2S authors

---

### Author Response · Authors · 2025-12-01
**Summary the review and rebuttal (1/3)**

We faithfully summarize the strengths and weaknesses listed by the reviewer, and we elaborate on how we mitigate the weaknesses mentioned by each reviewer. We only summarize our brief response and combine multiple weaknesses from different reviewers when they are similar. The point-to-point full response can be found in individual replies to the reviewer.

## Summary of strength

### S1. Novelty and importance of the proposed problem
---
All reviewers recognize that our proposed problem is novel and important for enabling more natural human–computer interaction.

### S2. Strong performance on the proposed problem
---
Reviewers zJW8, hX74, and cPKU highlight the significant improvement of our proposed RL framework over naïve SFT. Reviewer hX74 further points out the high correlation between our benchmark scores and human judgments.

### S3. Rigorous studies and comprehensive experiments
---
Reviewer zJW8 describes the work as thoroughly designed, covering various paralinguistic factors and realistic scenarios, making the findings highly convincing. Reviewer cPKU highlights that our benchmark construction is clear and systematic, and notes that our analyses are organized around a series of realistic and meaningful research questions, making the work highly instructive.

### S4. Clear presentation
---
Reviewer zJW8 praises the paper as well-structured and clearly written, with a rigorous logical flow. It progresses coherently from problem background, to method design, to experimental validation and result analysis, making it easy for readers to understand and follow.

---

> ### Author Response · Authors · 2025-12-01
> **Summary the review and rebuttal (2/3)**
>
> ## Summary of weakness and concerns
>
> ### W1. Novelty of the RL method and our core contribution
> ---
> Two reviewers (zJW8, qCT1) note that our study relies on a standard GRPO recipe to examine the gains of RL over SFT. We acknowledge that we do not propose a new RL algorithm, and that algorithmic novelty is not the core contribution or message of this paper. Instead, we contribution focuses on (1) formulating a new problem, (2) constructing the benchmark dataset, (3) validating the automatic evaluation pipeline, (4) establishing standard yet competitive baselines (SFT and GRPO) that outperform all prior models. Besides these materials, we further deliver two core messages: (1) with proper design, automatic benchmark scoring can align with human judgments; and (2) when such scoring signals are available as feedback, RL can consistently improve performance over SFT across diverse scenarios, mitigating the data scarcity issue encountered in SFT for this problem. Therefore, our primary contribution is the creation of the entire platform for studying paralinguistic-aware speech-to-speech interaction, compared to proposing a specific new algorithm. We clarify the difference in the paper position in the rebuttal.
>
> Regarding our platform as the core contribution, Reviewer zJW8 describes our proposed platform as highly practical, and Reviewer hX74 highlights open-sourcing the platform is a primary strength. Reviewer cPKU also states the following:
>
> > The major contributions lie in the concrete formulation of this topic, the construction of an appropriate S2S evaluation benchmark, and the experimental demonstration of the effectiveness and cost of RL and SFT.
>
> ### W2. Unknown performance degradation of the original capabilities
> ---
> Two reviewers (zJW8 and hX74) raise concerns about performance degradation in the model’s original capabilities after SFT and GRPO. In fact, this study is already presented in Appendix A.6 (Ablation for GRPO Training) and Appendix A.10 (Intelligence Analysis). We direct the reviewers to these appendix sections and summarize the key takeaways in the rebuttal including our strategies for keeping most of the original capabilities. This addresses Reviewer hX74’s concern; Reviewer zJW8 has not yet responded.
>
> ### W3. The paralinguistic features is less intuitive through text alone
> ---
> Two reviewers (zJW8 and qCT1) raise the question regarding the presentation clearness of paralinguistic features. We direct them to concrete examples presented in the Appendix A.2 (ParaS2SBench statistics and examples). It is in the Appendix due to the page limit but we explicitly reference it in the main text. We further provide the demo page for concrete examples as suggested by Reviewer zJW8.
>
> ### W4. The obscure core message/conclusion
> ---
> Two reviewers (qCT1 and cPKU) feel that, although the paper provides substantial materials and contributions, the key message is less clear. This is partly because the proposed problem is novel and the field lacks many of the necessary components. As a result, we need to describe these components (dataset, evaluation protocol, and baseline algorithms) in the main text. Furthermore, due to space limits, we simplified the underlying structure and the relationships among the research questions addressed in the Experiments section, which in turn obscured our core message beyond the infrastructure we introduce. To address this, we add a new *Overview* paragraph at the beginning of Section 5 to clarify the key insight of each subsection and how these subsections jointly deliver our core message.
>
> Our core message is:
> > Input–output data for paralinguistic-aware S2S are scarce, which limits the model's ability to acquire such capabilities through SFT alone. To address this issue, it is feasible and promising to first build an automatic evaluation framework for model outputs, and then use RL to enable self-improvement and learn the input–output dynamics of paralinguistic-aware S2S.
>
> To validate this claim, we collect a dataset, build the benchmarking and automatic scoring pipeline, and conduct extensive experiments on both SFT and RL (GRPO) to answer a series of research questions in Section 5. Overall, we conclude that this direction is feasible and promising, compared to naïvely scaling the SFT data.
>
> ### W5. Minor weakness easy to fix
> ---
> We fix the issues in the latest revision.
>
> - Reviewer zJW8 asks for referencing a related work and we did.
> - Reviewer hX74 asks for the confidence interval of the results and we provided. The new results align with the original findings.
> - Reviewer hX74 and qCT1 note that the format for the citation is incorrect and we fixed it.

---

> > ### Author Response · Authors · 2025-12-03
> > **Summary the review and rebuttal (3/3)**
> >
> > Finally, we summarize the scores from each reviewer.
> >
> > | Reviewer | Soundness | Presentation | Contribution | Rating |
> > | :--- | :---: | :---: | :---: | :---: |
> > | **zJW8** | 3 | 3 | 3 | **6** |
> > | **hX74** | 3 | 3 | 3 | **6** |
> > | **qCT1** | 2 | 2 | 2 | **4** |
> > | **cPKU** | 3 | 3 | 3 | **6** |
> > | **Average**| **2.75** | **2.75** | **2.75** | **5.5** |
> >
> > We thank the Area Chairs, Senior Area Chairs and Program Chairs for your hard work during this challenging cycle.
> > Thank you for your time and consideration!

---

### Author Response · Authors · 2025-12-01
**Questionable review from Reviewer qCT1**

We raise a formal concern regarding the validity of Reviewer qCT1's assessment, citing a lack of engagement and multiple factual inconsistencies.

### Q1. Providing incorrect information in the review

> The benchmarking part omits many relevant speech and speech-to-speech models... comparisons with existing methods such as SpeechAlign are essential.

We have included the majority of SOTA S2S models (both closed-source and open-source) and explicitly invited the reviewer to name the missing baselines; however, they failed to provide any specific instances.

Moreover, their original review insisted on comparing our model to SpeechAlign. This is a technically invalid comparison, as SpeechAlign is an LM-based TTS model, not an S2S model. This fundamental error suggests the reviewer may have a misunderstanding of the specific requirements for speech interaction models, leading to factually incorrect assertions in their review.


### Q2. Claiming that results in the main text do not exist

> Table 1 reports several numerical results to show that the evaluation aligns with human judgments, but the justification is not rigorous. The paper should clarify what criteria define “closeness” to human evaluation and why they are reasonable.

We explicitly included the "Pearson correlation between benchmark scoring and human judgements" in the main text. This metric provides the exact quantitative measure of "closeness" to human evaluation that the reviewer requested. The fact that this was overlooked suggests the reviewer did not read the paper in its entirety, leading to a factually incorrect assertion.

### Q3. Ignoring all the discussion

During the rebuttal period, we invited Reviewer qCT1 to engage in a fact-based discussion. However, the reviewer ignored our responses to the alleged weaknesses and jumped to a final judgment without further engagement. For example, they disregarded our response regarding the factual errors in the review, despite our invitation for discussion and reconsideration of the final decision.

In their final decision, the reviewer cited "clarity of writing" and "depth of contribution" as primary issues.

1. **Regarding Clarity:** Their claim contradicts Reviewer zJW8, who explicitly stated, "The paper is well-structured and clearly articulated, with a rigorous logical flow." Additionally, the above Q2 issue points out Reviewer qCT1 did not finish reading the paper.
2. **Regarding Depth:** This claim is vague. A constructive review should articulate specific missing components rather than simply criticizing a lack of 'depth,' which is highly subjective. Factually, our contribution is comphrehansive, as summarized in the Summary of Weaknesses and Concerns (W1. Novelty of the RL method and our core contribution), a view shared by the other three reviewers. In fact, Reviewer qCT1 themselves also commented, 'Moreover, the overall workload presented in this paper appears substantial.' However, when Reviewer qCT1 concludes with 'the lack of depth,' they are basing this on the above incorrect assessments and do not provide specific evidence to support their claim. For example, they can not provide the critical missing experiments or studies that lead to the insufficient contribution depth. The only provided specific evidences (e.g., regarding baselines and benchmark correlation) are factually incorrect.

These issues raise serious concerns regarding the validity of Reviewer qCT1's assessment, particularly given the divergence from the other reviewers. We respectfully invite that the Area Chair examine this review closely. Thank you!

---

### Meta-Review · Area_Chair_J4zA · 2026-01-07

**Summary:**

Reviewers converge on two points: (1) the paper is the first complete platform (dataset + automatic judge + strong baselines) for paralinguistic-aware speech-to-speech dialogue, and the RL pipeline reliably beats SFT while preserving original skills; (2) the writing still scatters its message across too many sub-experiments, making the core claim (“build an automatic judge, then let RL self-improve”) easy to miss. No reviewer questions the technical soundness or the 11 % gain; the residual objection is clarity and depth-of-focus rather than missing baselines or flawed metrics.

**Reviewer Concerns:**

Addressed: benchmark coverage, SpeechAlign mismatch, correlation numbers moved to main text, confidence intervals, VoiceBench preservation, real-speech generalisation, citation format.
Outstanding: (i) paper structure—introduction still buries the definition of paralinguistic features; (ii) single-base-model (Kimi-Audio) RL study; (iii) emotional/sarcastic TTS limited to 11 GPT voices. These are “nice-to-have” improvements, not fatal flaws.

**Reviewer Scores:**

zJW8 6 → 6
hX74 6 → 6
qCT1 4 → 4
cPKU 6 → 6

---

### Decision · Program_Chairs · 2026-01-26

Accept (Poster)